# The sugar-responsive enteroendocrine neuropeptide F regulates lipid metabolism through glucagon-like and insulin-like hormones in *Drosophila melanogaster*

Yuto Yoshinari[1,2], Hina Kosakamoto [3,4], Takumi Kamiyama[2], Ryo Hoshino[2], Rena Matsuoka[2], Shu Kondo[5], Hiromu Tanimoto[6], Akira Nakamura [7,8], Fumiaki Obata [1,3,4,9,10] & Ryusuke Niwa [1,11✉]

The enteroendocrine cell (EEC)-derived incretins play a pivotal role in regulating the secretion of glucagon and insulins in mammals. Although glucagon-like and insulin-like hormones have been found across animal phyla, incretin-like EEC-derived hormones have not yet been characterised in invertebrates. Here, we show that the midgut-derived hormone, neuropeptide F (NPF), acts as the sugar-responsive, incretin-like hormone in the fruit fly, *Drosophila melanogaster*. Secreted NPF is received by NPF receptor in the corpora cardiaca and in insulin-producing cells. NPF-NPFR signalling resulted in the suppression of the glucagon-like hormone production and the enhancement of the insulin-like peptide secretion, eventually promoting lipid anabolism. Similar to the loss of incretin function in mammals, loss of midgut NPF led to significant metabolic dysfunction, accompanied by lipodystrophy, hyperphagia, and hypoglycaemia. These results suggest that enteroendocrine hormones regulate sugar-dependent metabolism through glucagon-like and insulin-like hormones not only in mammals but also in insects.

[1] Life Science Center for Survival Dynamics, Tsukuba Advanced Research Alliance (TARA), University of Tsukuba, Tsukuba, Ibaraki, Japan. [2] Graduate School of Life and Environmental Sciences, University of Tsukuba, Tsukuba, Ibaraki, Japan. [3] Department of Genetics, Graduate School of Pharmaceutical Sciences, The University of Tokyo, Bunkyo-ku Tokyo, Japan. [4] Laboratory for Nutritional Biology, RIKEN Center for Biosystems Dynamics Research, Kobe, Hyogo, Japan. [5] Genetic Strains Research Center, National Institute of Genetics, Mishima Shizuoka, Japan. [6] Graduate School of Life Sciences, Tohoku University, Sendai, Miyagi, Japan. [7] Graduate School of Pharmaceutical Sciences, Kumamoto University, Kumamoto, Japan. [8] Laboratory of Germline Development, Institute of Molecular Embryology and Genetics, Kumamoto University, Kumamoto, Japan. [9] Laboratory of Molecular Cell Biology and Development, Graduate School of Biostudies, Kyoto University, Kyoto, Japan. [10] AMED-PRIME, Japan Agency for Medical Research and Development Chiyoda-ku, Tokyo, Japan. [11] AMED-CREST, Japan Agency for Medical Research and Development, Chiyoda-ku, Tokyo, Japan. ✉email: ryusuke-niwa@tara.tsukuba.ac.jp

All organisms must maintain energy homoeostasis in response to nutrient availability. To maintain balance of catabolism and anabolism, organisms coordinate systemic energy homoeostasis through humoral factors. Insulin and counter-regulatory hormones, such as glucagon, have previously been shown to act as such humoral factors in response to nutritional and environmental cues[1–4]. Insulin promotes circulating carbohydrate clearance, while counter-regulatory hormones increase carbohydrate release into circulation. To date, much has been learned about how impaired insulin and/or counter-regulatory hormone actions contribute to carbohydrate metabolic dysregulation.

In addition to the glucagon- and insulin-secreting pancreatic cells, the intestine is also a key to regulating systemic energy homoeostasis. Especially, enteroendocrine cells (EECs) secrete multiple hormones to orchestrate systemic metabolic adaptation across tissues[5–8]. Recent works have revealed that EECs sense multiple dietary nutrients and microbiota-derived metabolites that influence the production and/or secretion of enteroendocrine hormones[7–12]. In mammals, an enteroendocrine hormone that stimulates the secretion of glucagon and insulin, particularly the latter, is referred to as "incretin", such as glucose-dependent insulinotropic polypeptide (GIP) and glucagon-like pepetide-1 (GLP-1)[5]. The secretion of GIP and GLP-1 is stimulated by dietary carbohydrates and lipids. Incretins stimulate pancreatic insulin secretion and conversely suppress glucagon secretion in a glucose-dependent manner. The physiological importance of incretins is epitomised by the fact that dysregulation of incretins often associates with obesity and type 2 diabetes[6,13].

To further dissect the molecular, cellular, and endocrinological mechanisms of glucagon and insulin actions in animals, the fruit fly, *Drosophila melanogaster* has emerged as a powerful genetic system in recent years. There are eight genes encoding *Drosophila* insulin-like peptides (DILPs), designated DILP1 to DILP8. Among these DILPs, it is thought that DILP2, DILP3, and DILP5 are particularly essential for the regulation of haemolymph glucose levels and fat storage, controlling developmental timing, body size, and longevity[14–16]. *D. melanogaster* also possesses a hormone that is functionally equivalent to the mammalian glucagon, called adipokinetic hormone (AKH). AKH is produced in and secreted from a specialised endocrine organ, the corpora cardiaca (CC), and acts on the fat body, leading to lipolysis-dependent energy metabolism. Furthermore, recent studies have identified two factors secreted by EECs, Activin-β and Bursicon-α (Bursα), which play essential roles in modulating AKH-dependent lipid metabolism in the fat body[9,11]. However, neither Activin-β nor Bursα directly acts on the CC or insulin-producing cells (IPCs). Indeed, no incretin-like enteroendocrine hormones has been discovered in invertebrates.

Here, we report that the midgut-derived hormone neuropeptide F (NPF), a homologue of the mammalian neuropeptide Y (NPY), acts as the sugar-responsive, incretin-like hormone in *D. melanogaster*, while the primary structure of NPF is completely different from that of GIP or GLP-1. NPF is produced in and secreted from midgut EECs in response to dietary nutrients. NPF is bound by NPF receptor (NPFR) that is present in the CC and IPCs. Impairment of NPF/NPFR signalling resulted in AKH- and insulin-dependent catabolic phenotypes, accompanied by hypoglycaemia, lipodystrophy, and hyperphagia. Our work demonstrates a key role of inter-organ communication between the midgut, the brain and endocrine organs to regulate energy homoeostasis.

## Results

**Midgut NPF is required for lipid accumulation in the fat body and promotes starvation resistance**. We have previously reported that midgut-derived NPF is essential for mating-induced germline stem cell increase in female *D. melanogaster*[17].

This discovery prompted us to ask whether midgut-derived NPF is also involved in other biological processes. In particular, since many enteroendocrine hormones are known to regulate nutritional plasticity[9–11,18], we inquired whether loss of midgut-derived NPF leads to any nutrient-related phenotypes. To knock down *NPF* specifically in EECs, we utilised *TKg-GAL4*. This *GAL4* driver is active in most *NPF+* EECs[17,18] and small subsets of neurons but not in *NPF+* neurons[17]. *NPF* knockdown with *TKg-GAL4* (*TKg>NPF^RNAi*) successfully reduced the number of *NPF+* EECs and *NPF* mRNA expression in the midgut (Supplementary Fig. 1a, b), as previously reported[17]. We found that the flies became significantly sensitive to nutrient deprivation. Adult flies were raised on normal food for 6 days after eclosion, and then transferred to a 1% agar-only medium. *TKg>NPF^RNAi* animals showed hypersensitivity to nutrient deprivation compared to control animals (*TKg>LacZ^RNAi*) (Fig. 1a). The hypersensitivity was observed with two independent *UAS-NPF^RNAi* constructs (KK and TRiP; see Methods), each of which targeted a different region of the *NPF* mRNA.

A recent study has reported that the loss-of-function of another midgut-derived peptide hormone, Bursα also exhibited hypersensitivity to starvation[11]. We examined whether the *NPF* loss-of-function phenotype was due to the expression and/or secretion defect in Bursα in the gut. However, *NPF* knockdown in the EECs did not affect *Bursα* mRNA expression in the intestine or Bursα accumulation in the EECs of the posterior midgut (Supplementary Fig. 1b, c).

The survivability of flies on nutrient deprivation directly correlates with accessibility to energy storage in their bodies, mainly stored as neutral lipids, including triacylglycerides (TAG) in the fat body[19,20]. Consistent with the starvation hypersensitivity in animals with loss of *NPF* function, we detected a significant overall reduction of whole-body TAG levels in both *TKg>NPF^RNAi*-animals and *NPF* genetic null mutants (*NPF^sk1/Df*) (Fig. 1b, Supplementary Fig. 1e). Further, in the fat body of both *TKg>NPF^RNAi* animals and *NPF* mutants, the signal intensity of the lipophilic fluorescent dye (LipidTOX) was significantly reduced, as compared with control animals (Fig. 1c; Supplementary Fig. 1f). Conversely, overexpression of *NPF* in the EECs resulted in a slight increase in TAG abundance (Supplementary Fig. 1g).

In addition to the RNAi animals, we found that *NPF* genetic null mutants (*NPF^sk1/Df*) also exhibited similar hypersensitivity phenotype on starvation (Supplementary Fig. 1d). Importantly, transgenic *NPF* reintroduction into EECs (*TKg>NPF; NPF^sk1/Df*) was sufficient to recover hypersensitivity to starvation and the TAG reduction observed in *NPF* mutant background (Fig. 1d–f). These results suggest that NPF from midgut EECs is required to sustain organismal survival during nutrient deprivation.

To rule out the possibility that loss of *NPF* during the larval and pupal stages impacts adult metabolism, we conducted adult-specific knockdown of *NPF* with *tub-GAL80^ts* (*TKg^ts>NPF^RNAi*). In *TKg^ts>NPF^RNAi* adults, a temperature-shift to restrictive temperatures following eclosion significantly reduced NPF levels in EECs (Supplementary Fig. 2a). Moreover, the adult-specific knockdown of *NPF* resulted in hypersensitivity upon starvation and reduced TAG abundance (Fig. 1g–i), while no visible alterations were noted in size or morphology of the fat body (Fig. 1i). We also observed a significant reduction in circulating glucose and trehalose levels in *TKg^ts>NPF^RNAi* adults at restrictive temperature (Fig. 1j, Supplementary Fig. 1h, 2b), suggesting that reduced lipid storage results in high utilisation of circulating glucose.

Since energy storage well correlates with the amount of food consumption, the lean phenotype described above may be simply due to less food intake. However, a CAFÉ assay[21] revealed that both *TKg>NPF^RNAi* animals and *NPF* mutants increased food intake

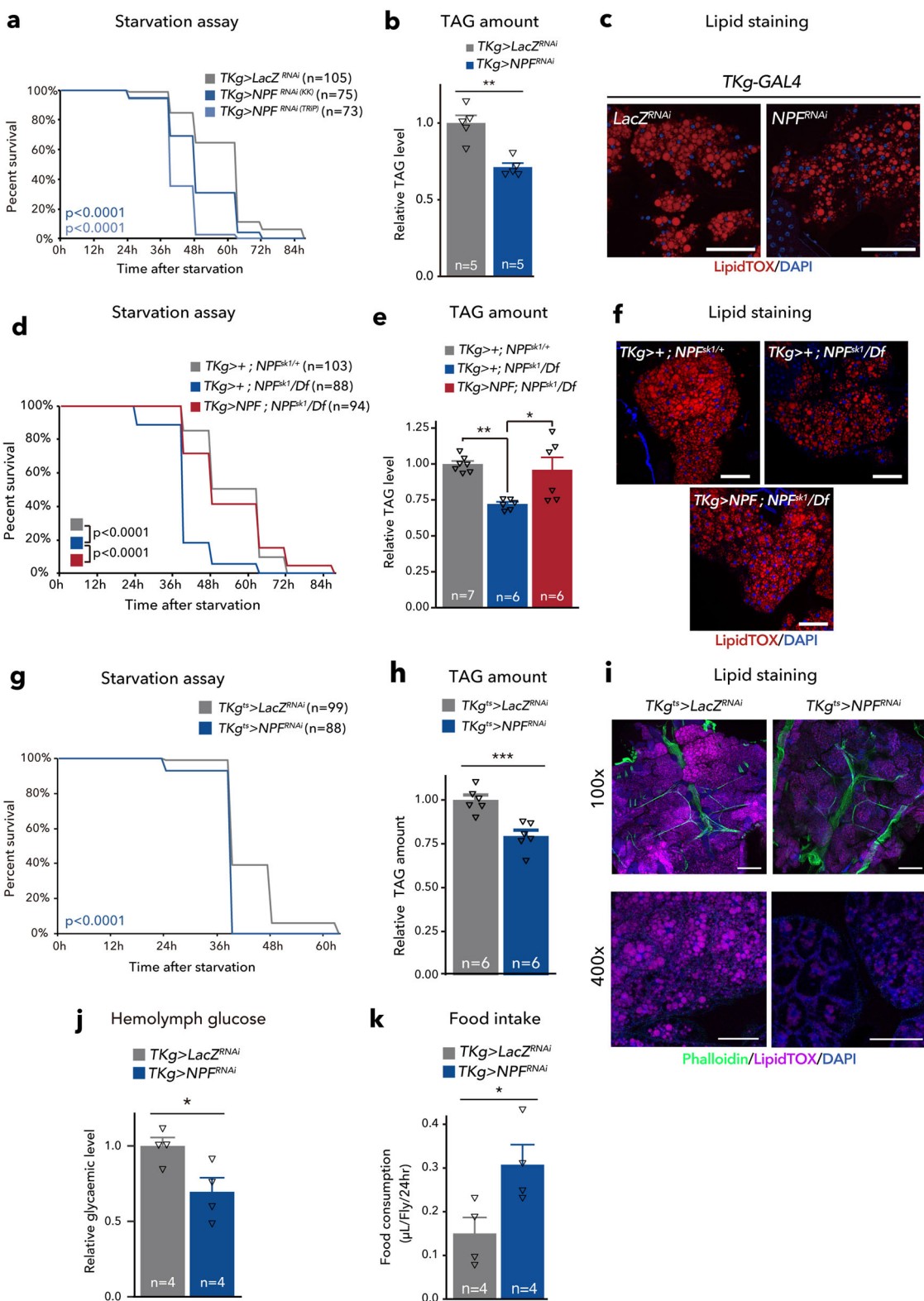

(Fig. 1k; Supplementary Fig. 1i, 2c). Therefore, the hypersensitivity to starvation and the lean phenotype of animals with loss of *NPF* function do not seem to be secondary to food intake defects, but a more direct outcome of some metabolic defects.

**Brain NPF is not involved in lipid accumulation in the fat body or the promotion of starvation resistance.** It is well known that NPF produced in the brain has orexigenic function[22,23].

Therefore, high food intake in *TKg>NPF^RNAi* suggests opposing functions between the brain-derived and midgut-derived NPF. To examine whether brain NPF affects lipid metabolism, we employed *fbp-GAL4*[24], which is active in the NPF[+] neurons in the brain, but not in gut EECs (Supplementary Fig. 3a). Knockdown of *NPF* with *fbp-GAL4* (*fbp>NPF^RNAi*) abolished anti-NPF antibody immunoreactivity in two sets of large neurons, termed L1-l and P1[25], in the brain without affecting NPF level in the gut

**Fig. 1 NPF from midgut EECs maintains metabolic homoeostasis. a–i** Phenotypes of the midgut EEC-specific *NPF* knockdown animals (*TKg>NPF^RNAi*) (a-c), *NPF* genetic mutant animals with or without midgut-specific *NPF* reintroduction (*TKg>NPF*; *NPF^sk1/Df*) (**d-f**), and adult EEC-specific *NPF* knockdown animals (*TKg^ts>NPF^RNAi*) (**g-i**). **a, d, g** Survival during starvation. **b, e, h** Relative TAG amount. **c, f, i** LipidTOX (red or magenta) and DAPI (blue) staining of dissected fat body tissue. Scale bar, 50 μm in **c** and **f**, 200 μm (100×) and 50 μm (400×) in **i**. **j** Relative circulating glucose levels. **k** Feeding quantity measurement with CAFÉ assay. For RNAi experiments, *LacZ* knockdown (*TKg>LacZ^RNAi*) was used as the negative control. For all bar graphs, the number of samples assessed (*n*) is indicated in each graph. Mean ± SEM with all data points is shown. Statistics: Log rank test with Holm's correction (**a, d**, and **g**), two-tailed Student's *t*-test (**b, h, j**, and **k**), one-way ANOVA followed by Tukey's multiple comparisons test (**e**). *$p < 0.05$, **$p < 0.01$. *p*-values: **a** $p < 0.0001$ (*TKg>LacZ^RNAi* vs. *TKg>NPF^RNAiTRiP*), $p < 0.0001$ (*TKg>LacZ^RNAi* vs. *TKg>NPF^RNAiKK*); **b** $p = 0.0005$, **d** $p < 0.0001$ (*TKg>+; NPF^sk1/+* vs. *TKg>+; NPF^sk1/ NPF^Df*), $p < 0.0001$ (*TKg>+; NPF^sk1/ NPF^Df* vs. *TKg>NPF; NPF^sk1/NPF^Df*); **e** $p = 0.0027$ (*TKg>+; NPF^sk1/+* vs. *TKg>+; NPF^sk1/NPF^Df*), $p = 0.0112$ (*TKg>+; NPF^sk1/ NPF^Df* vs. *TKg>NPF; NPF^sk1/NPF^Df*); **g** $p < 0.0001$; **h** $p = 0.0008$; **j** $p = 0.0316$; **k** $p = 0.0363$.

(Supplementary Fig. 3b). In *fbp>NPF^RNA* adults, a mild reduction in food consumption was observed without impacting starvation resistance or TAG abundance (Supplementary Fig. 3c-e). Moreover, reintroduction of *NPF* in the brain (*fbp>NPF*; *NPF^sk1/Df*) did not recover the metabolic phenotypes of the *NPF* mutant (Supplementary Fig. 3f-g). These results contrast those obtained following the reintroduction of *NPF* in the midgut (*TKg>NPF*; *NPF^sk1/Df*; Fig. 1d, e). Collectively, these results suggest that midgut NPF has a prominent role in suppressing lipodystrophy, which is independent from the brain NPF.

**Midgut NPF is required for energy homoeostasis.** To further explore the lean phenotype of *TKg>NPF^RNAi* animals at the molecular level, we conducted an RNA-seq transcriptome analysis on the abdomens of adult females. Among the 105 curated carbohydrate metabolic genes, 17 were significantly upregulated in *TKg>NPF^RNAi* animals ($p < 0.05$; Supplementary Fig. 4a, Supplementary Data 1). Many of these genes were also upregulated in *TKg>NPF^RNAi* samples, however, these results were not statistically significant because replicate No. 1 of *TKg>LacZ^RNAi* exhibited deviation in the expression pattern (Supplementary Fig. 4a, Supplementary Data 1). Moreover, among the 174 curated genes involved in mitochondrial activity and genes encoding electron respiratory chain complexes, 53 were significantly upregulated ($p < 0.05$) in *TKg>NPF^RNAi* samples (Supplementary Fig. 4b, Supplementary Data 2). Metabolomic analysis demonstrated a significant shift in the whole-body metabolome of *TKg>NPF^RNAi* animals (Fig. 2a, Supplementary Fig. 5a, Supplementary Data 3, 4). We found that, while circulating glucose level in the haemolymph was significantly decreased (Fig. 1g), *TKg>NPF^RNAi* resulted in increase of tricarboxylic acid (TCA) cycle metabolites, such as citrate, isocitrate, fumarate, and malate, in whole-body samples as well as haemolymph samples (Fig. 2b, c). These data strongly suggest that *TKg>NPF^RNAi* animals utilise and direct more glucose into the TCA cycle.

Based on RNA-seq transcriptome analysis, we found that starvation-induced genes[19] were also upregulated in the abdomens of *TKg>NPF^RNAi* adults (Fig. 2d, Supplementary Data 5). Subsequent quantitative PCR (qPCR) validated the upregulation of the starvation-induced gluconeogenetic genes (*fructose-1,6-bisphosphatase* (*fbp*) and *Phosphoenolpyruvate carboxykinase 1* (*pepck1*))[26] (Fig. 2e). In general, TAG is broken into free fatty acids to generate acetyl-coenzyme A (CoA), which is metabolised in the mitochondria through the TCA cycle and oxidative phosphorylation. We also confirmed the upregulation of lipid metabolism gene (*Brummer* (*Bmm*)) in the abdomen of *TKg>NPF^RNAi* animals (Fig. 2f). Notably, upregulation of *Acetyl-CoA carboxylase* (*ACC*) was not reproduced with qPCR (Fig. 2f). These data suggest that *TKg>NPF^RNAi* animals are in the starved-like status despite taking in more food, and that haemolymph glucose levels cannot be maintained even with the activation of gluconeogenesis and lipolysis in *TKg>NPF^RNAi* animals. We hypothesise that, owing to the starved-like status,

the loss of midgut *NPF* function might lead to an abnormal consumption of TAG, resulting in the lean phenotype.

**Midgut NPF responds to dietary sugar.** Since EECs can sense dietary nutrients, we surmised that dietary nutrients affect NPF production and/or secretion in midgut EECs. We thus compared NPF protein and mRNA levels in flies fed standard food or starved for 48 h with 1% agar. After 48 h of starvation, NPF protein in midgut EECs was significantly increased (Fig. 3a, b), although its transcript in the intestine was reduced (Fig. 3c). These data suggest that the increased accumulation of NPF protein in EECs upon starvation is not due to upregulation of *NPF* mRNA expression level, but rather due to post-transcriptional regulation. This situation was very similar to the case of mating-dependent change of NPF protein level, and may reflect the secretion of NPF protein from EECs[17]. Considering that the high accumulation of NPF protein without *NPF* mRNA increase indicate a failure of NPF secretion, we hypothesised that starvation suppresses NPF secretion from EECs.

To identify specific dietary nutrients that affect NPF levels in EECs, after starvation, we fed flies a sucrose or Bacto peptone diet as exclusive sources of sugar and proteins, respectively. Interestingly, by supplying sucrose, the levels of both of NPF protein and *NPF* mRNA in the gut reverted to the levels similar to ad libitum feeding conditions (Fig. 3a, b). In contrast, Bacto peptone administration did not reduce middle midgut NPF protein level, but rather increased both NPF protein and *NPF* mRNA levels (Fig. 3c). These data imply that midgut NPF is secreted primarily in response to dietary sugar, but not proteins. This sucrose-dependent NPF secretion was observed in flies fed a sucrose medium for 6 h after starvation, whereas a 1h sucrose restoration had no effect on NPF accumulation (Supplementary Fig. 6a).

**Sugar-responsive midgut NPF production is regulated by the sugar transporter Sut1.** In mammals, the sugar-stimulated secretion of GLP-1 is partly regulated by glucose transporter 2, which belongs to the low-affinity glucose transporter solute carrier family 2 member 2 (SLC2)[27,28]. In *D. melanogaster*, a SLC2 protein, Glucose transporter 1 (Glut1), in the Bursα+ EECs regulates sugar-responsible secretion and *Bursα* mRNA expression[11]. However, knockdown of *Glut1* did not affect *NPF* mRNA nor NPF protein abundance in EECs (Supplementary Fig. 6b, c). Thus, we next examined which SLC2 protein, aside from Glut1, regulates NPF levels in the gut. There are over 30 putative homologues of *SLC2* in the *D. melanogaster* genome[29]. Of these, we focused on *sugar transporter1* (*sut1*), because its expression has been described in the intestinal EECs by FlyGut-seq project[30] and Flygut EEs single-cell RNA-seq project[31]. To verify *sut1* expression, we generated a *sut1^Knock-in(KI)-T2A-GAL4* strain using CRISPR/Cas9-mediated homologous recombination[32,33]. Consistent with these transcriptomic analyses, *sut1^KI-T2A-GAL4* expression was observed in the EECs, including NPF+ EECs

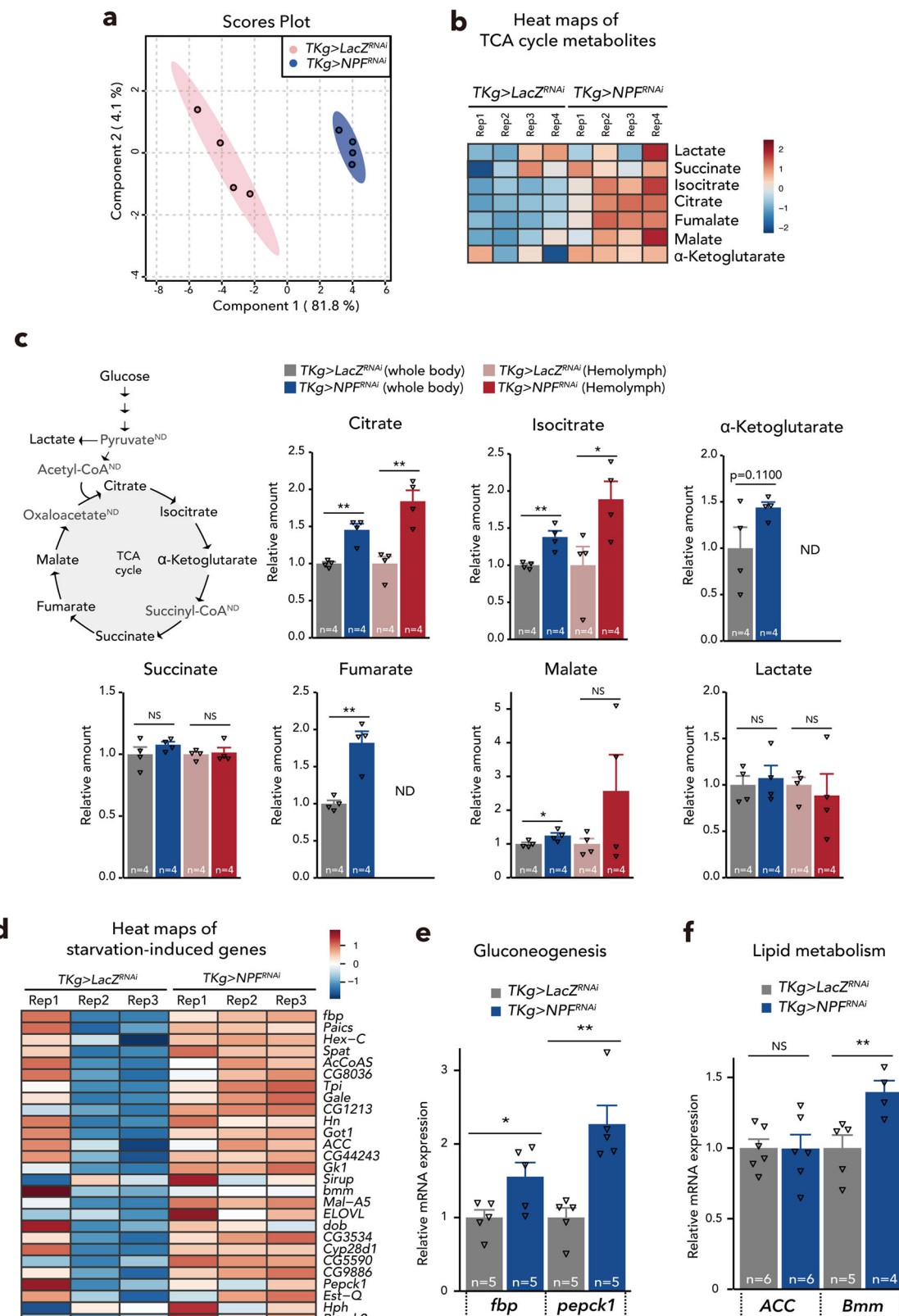

(Fig. 3d). In addition, we found that overexpressed monomeric Venus (mVenus)-tagged Sut1 protein (*TKg>sut1::mVenus*) was localised on the membrane of NPF[+] EECs (Fig. 3e), supporting the notion that Sut1 mediates the transport of extracellular sugar.

Next, to ascertain the glucose transporter capacity of Sut1, we expressed *FLII[12]Pglu-700μδ6*, a fluorescence resonance energy transfer (FRET)-based glucose sensor (referred to as Glu[700])[34,35], into *D. melanogaster* S2 cells, with or without *sut1* over-expression. In this experiment, we equilibrated S2 cells in buffer lacking glucose[35], followed by application of high-glucose (25 mM final concentration) solution. As compared with S2 cells without *sut1* overexpression, the addition of high-glucose

**Fig. 2 Midgut-derived NPF regulates systemic carbohydrate/lipid metabolism. a** Principal component analysis plot of metabolome data as in Supplementary Fig. 5. Note that *NPF* knockdown animals indicate dispersed cluster with control animals. **b** Heat maps of selected metabolites in whole-body samples. Red and blue indicate increased and decreased metabolites relative to median metabolite levels of *TKg>LacZ*$^{RNAi}$, respectively; the ratios were plotted on a colour scale (right). **c** LC–MS/MS measurement of whole-body or haemolymph metabolites in control and *TKg>NPF*$^{RNAi}$. ND; no data. The number of samples assessed (*n*) is indicated in each graph. **d** Expression heatmap of a curated set of starvation-induced genes. Red and blue indicate increased and decreased gene expressions relative to median gene expression levels of *TKg>LacZ*$^{RNAi}$, respectively; the ratios were plotted on a colour scale (upper right corner). Gene expression levels are represented by TMM-normalised FPKM. **e** and **f** Relative fold change for starvation-induced genes (**e**: *fbp* and *pepck*, **f**: *ACC* and *Bmm*) in dissected abdomens of female adult control and *TKg>NPF*$^{RNAi}$ animals, as determined by qPCR. Samples are normalised to *rp49*. The number of samples assessed (*n*) is indicated in each graph. For RNAi experiments, *LacZ* knockdown (*TKg>LacZ*$^{RNAi}$) was used as the negative control. For all bar graphs, mean ± SEM with all data points is shown. Statistics: two-tailed Student's *t*-test (**c, e,** and **f**). *$p < 0.05$, **$p < 0.01$; NS, non-significant ($p > 0.05$). *p*-values: **c** (Citrate) Whole Body (WB), $p = 0.0019$, Haemolymph (Hm), $p = 0.0031$, (Isocitrate) WB, $p = 0.0048$, Hm, $p = 0.0436$, (α-Ketoglutarate) WB, $p = 0.1101$, (Succinate) WB, $p = 0.2639$, Hm, $p = 0.7650$, (Fumarate) WB, $p = 0.0023$ (Malate) WB, $p = 0.0403$, Hm, $p = 0.2410$, (Lactate) WB, $p = 0.6794$, Hm, $p = 0.6588$; **e** $p = 0.0349$ (*fbp*), $p = 0.0022$ (*pepck1*); **f** $p = 0.9753$ (*ACC*), $p = 0.0165$ (*Bmm*).

solution significantly elevated the *Glu*$^{700}$ FRET signal with *sut1* overexpression (Supplementary Fig. 7a). We also investigated whether *sut1* regulates cellular glucose levels in EECs with a *UAS-Glu*$^{700}$ transgenic strain. Knockdown of *sut1* in the EECs caused a slight, but significant, decrease in the *Glu*$^{700}$ FRET signal (Supplementary Fig. 7b), whereas 24 h of starvation caused a more significant reduction in FRET signal intensity. Taken together, these data suggest that Sut1 regulates intracellular glucose levels and may transport glucose into cells.

We next examined the effect of knockdown of *sut1* in the EECs. The *sut1* knockdown with a transgenic RNAi lines (*TKg>sut1*$^{RNAiKK}$) resulted in the decrease in *NPF* mRNA level in the midgut, similar to what we observed in starvation conditions (Fig. 3f). On the other hand, *sut1* knockdown resulted in the increase in NPF protein level in EECs in ad libitum feeding condition, while there was no significant difference in NPF protein level in starvation condition, compared with control (Supplementary Fig. 6d, e). Moreover, *sut1* knockdown disrupted the reversion of NPF accumulation by sucrose restoration (Supplementary Fig. 6d, e). *NPF* mRNA expression was also significantly reduced with an trend of increase in NPF protein abundance, in another transgenic RNAi animal model (*TKg>sut1*$^{RNAiTRiP}$), and *sut1* null mutant animals generated by CRISPR/Cas9 system[36] (Fig. 3g, Supplementary Fig. 8a–f). Consistent with the NPF accumulation phenotype, *sut1* knockdown (both *TKg>sut1*$^{RNAiKK}$ and *TKg>sut1*$^{RNAiTRiP}$) resulted in hypersensitivity to starvation and reduction in lipid amount (Fig. 3h–j, Supplementary Fig. 8c, d). Importantly, brain-specific *sut1* knockdown using *Otd-FLP* did not cause NPF accumulation in the midgut, while it did slightly reduce the abundance of TAG (Supplementary Fig. 9a-c). Moreover, *sut1*$^{KI-T2A}$-*GAL4* was not expressed in NPF$^{+}$ neurons in the brain (Supplementary Fig. 9d), suggesting that brain *sut1* is not involved in the regulation of midgut NPF production or secretion. Furthermore, *sut1* knockdown did not reduce *Bursα* mRNA expression in the gut (Supplementary Fig. 9e). These data suggest that Sut1 in the EECs is indispensable for midgut NPF production and whole animal lipid metabolism.

**NPFR in the CC regulates lipid metabolism.** We have previously reported that midgut EEC-derived NPF may be secreted into circulation and activate NPFR in the ovarian somatic cells, leading to germline stem cell proliferation[17]. We first investigated potential NPF-dependent lipid metabolism regulation by ovarian NPFR. However, *NPFR* knockdown in the ovarian somatic cells with *Traffic jam(tj)-GAL4* did not induce hypersensitivity to starvation or reduction of TAG contents (Supplementary Fig. 10a, b), implying that *NPFR* expressed in tissues other than the ovary must be involved in regulating sugar-dependent lipid metabolism.

To determine the tissues expressing *NPFR*, we utilised two independent *NPFR* knock-in *T2A-GAL4* lines, *NPFR*$^{KI-T2A}$-*GAL4* (see the "Methods" section) and *NPFR*$^{KI-RA/C}$-*GAL4*[37], each of which carry a transgene cassette that contained *T2A-GAL4*[38] immediately in front of the stop codon of the endogenous *NPFR* gene. Crossing these lines with a *UAS-GFP* line revealed GFP expression not only in the brain (Supplementary Fig. 11a), as previously reported[37], but also in other tissues, including the CC (Fig. 4a, Supplementary Fig. 11b), short neuropeptide F (sNPF)$^{+}$ enteric neurons, Malpighian tubules, ovary, and gut (Supplementary Fig. 11c–f). The expression in the CC was observed in two independent *KI-GAL4* lines, *NPFR*$^{KI-T2A}$-*GAL4* and *NPFR*$^{KI-RA/C}$-*GAL4* (Fig. 4a, Supplementary Fig. 11b). Therefore, based on these results and those of a previous RNA-seq analysis[39], we surmised that *NPFR* is expressed in the CC. Since the CC produces the glucagon-like peptide, AKH, which regulates organismal carbohydrate and triglyceride metabolism in insects[2,20,40–42], we were particularly interested in examining whether NPFR in the CC is involved in metabolic regulation in adult *D. melanogaster*.

To this end, we further conducted starvation experiments. Similar to animals with loss of *NPF* function, *NPFR* knockdown animals (*Akh>NPFR*$^{RNAiKK}$ or *Akh>NPFR*$^{RNAiTRiP}$) and *NPFR*-null mutants (*NPFR*$^{sk8}$/*Df*) were more sensitive to starvation, compared with control (*Akh>LacZ*$^{RNAi}$, *NPFP*$^{sk8}$/+, or *Df*/+) (Fig. 4b; Supplementary Fig. 10c). The *NPFR* knockdown animals exhibited reduction of TAG amount and glycaemic levels, accompanied by increase of food intake, similar to animals with disrupted *NPF* (Fig. 4c–f; Supplementary Fig. 10d). Moreover, reintroduction of *NPFR* in the CC rescued the starvation sensitivity, the low TAG levels, and the reduced signal intensity of the LipidTOX in the fat body of *NPFR* mutants (Fig. 4g–i), indicating that NPFR in the CC is essential for modulating lipid catabolism.

Consistent with a previous report[31], *NPFR*$^{KI-T2A}$-*GAL4* was also expressed in the visceral muscles (Supplementary Fig. 11f). We therefore knocked down *NPFR* in the visceral muscle with *how-GAL4*, a genetic driver active in the visceral muscle. In the adult females of this genotype, TAG amount was reduced, but hypersensitivity to starvation was not observed (Supplementary Fig. 12a, b). Therefore, we conclude that NPFR in the CC has a pivotal role in lipid metabolism coupled with its role in starvation resistance.

**NPF/NPFR signalling controls glucagon-like hormone production.** Consistent with the attenuation of lipid catabolism by NPF and NPFR, *Akh* mRNA level was significantly upregulated in midgut EEC-specific *NPF* knockdown or CC-specific *NPFR* knockdown (Fig. 5a). Furthermore, AKH protein levels in the CC were significantly reduced in *NPF* and *NPFR* knockdown animals (Fig. 5b, c). Given that these phenotypes resembled the excessive AKH

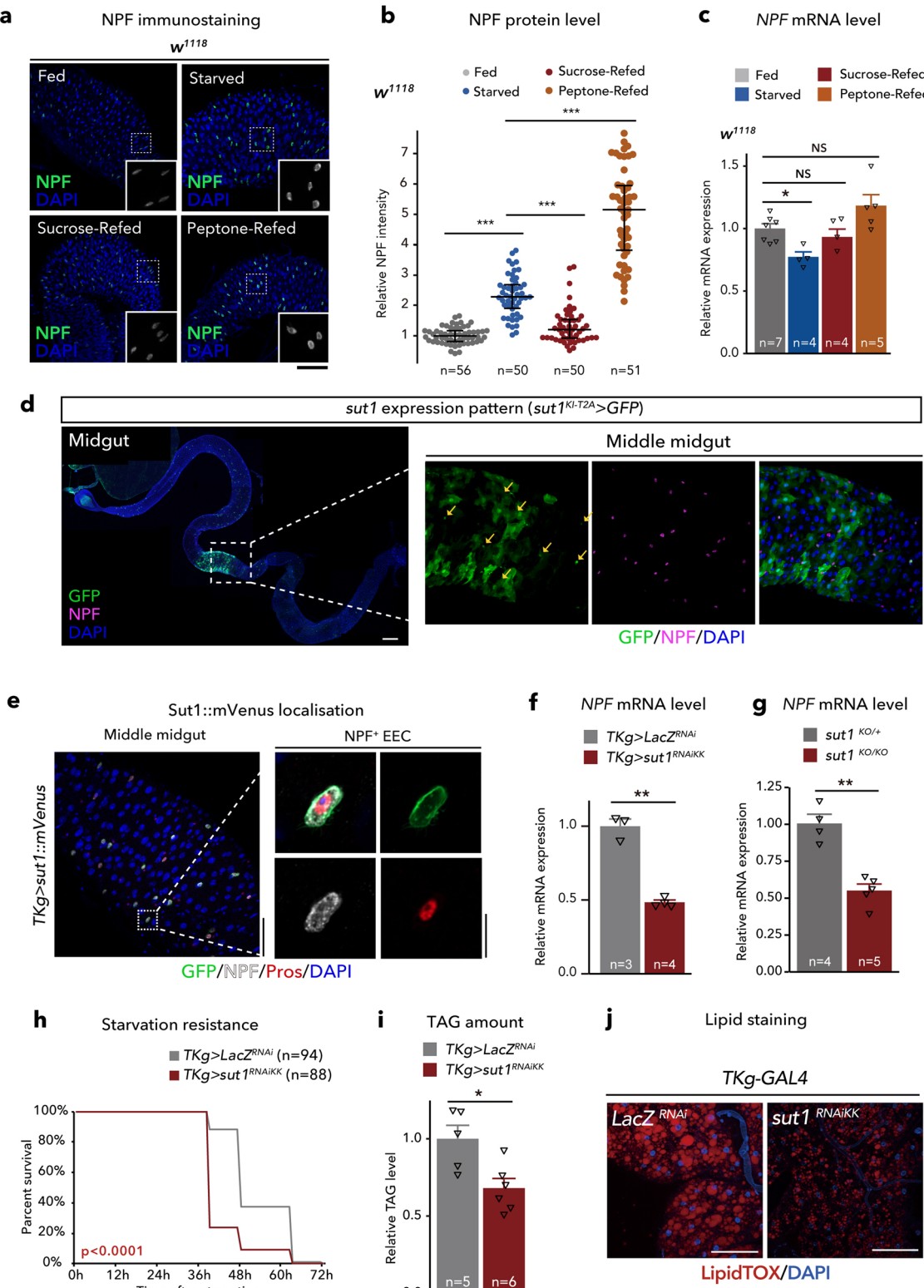

secretion reported in a previous study[43], these results suggest that upregulation of AKH production and secretion induces the metabolic phenotype of loss of *NPF* or *NPFR* function animals. To test this hypothesis, we assessed a relationship between NPF–NPFR signalling and AKH–AKH receptor (AKHR) signalling in lipid metabolism. The reduction of TAG and starvation hypersensitivity of *NPFR* knockdown were rescued by a CC-specific co-suppression of *Akh* (*Akh>NPFR^RNAi+AkhRNAi*) (Fig. 5d–f). Further, the low

TAG level and starvation sensitivity of *NPF* knockdown was also rescued with *Akh^KO* mutants (Supplementary Fig. 13a, b), suggesting that lipodystrophy of NPF/NPFR-deficient animals is mediated by AKH. Notably, the knockdown of *Akh* alone (*Akh>Akh^RNAi*) resulted in high starvation resistance and increased TAG abundance compared to *Akh>NPFR^RNAi+Akh^RNAi* (Fig. 5d, f), implying that other factor(s) from the CC may contribute to the lipid storage reduction in *NPFR* knockdown animals. Complementary to these

**Fig. 3 NPF in midgut EECs is regulated by dietary nutrients. a** Immunostaining for NPF (green/white) and DAPI (blue) in the adult middle midguts collected from 6-day-old control ($w^{1118}$) animals fully fed, 48 h starved animals, and animals re-fed with sucrose or Bacto peptone following 24 h starvation (sucrose-refed/peptone-refed). Scale bar, 50 μm. **b** Quantifications of NPF fluorescent intensity under the conditions described for (**a**). The number of EECs (*n*) analysed in each genotype is indicated in the graph. Each point represents NPF fluorescent intensity in a single EEC. For each genotype, more than eight guts were used. **c** RT-qPCR analysis of *NPF* mRNA level under the conditions described for (**a**). **d** Immunofluorescence of *sut1^KI-T2A*-GAL4-driven *UAS-GFP* (*sut1^KI-T2A*>*GFP*) in the midgut. The sample was co-stained with anti-NPF antibody (magenta) and DAPI (blue). The GFP signal (green) was detected in NPF+ EECs (arrows). Of note, *sut1^KI-T2A*-GAL4 driven GFP signal were visible in ~40% of NPF+ EECs. Scale bar, 50 μm. **e** Immunofluorescence of *TKg-GAL4*-driven *UAS-sut1::mVenus* (*TKg*>*sut1::mVenus*) in the midgut. The sample was co-stained with anti-NPF antibody (white), anti-Prospero antibody (red), and DAPI (blue). Sut1::mVenus (green) is localised on the cell membrane of EECs. Scale bar, (left) 50 μm, (right) 10 μm. **f, g** RT-qPCR analysis of *NPF* mRNA level in EEC-specific *sut1* knockdown (*TKg*>*sut1^RNAi*) (**f**) and *sut1* genetic mutant (**g**) animals. **h** Survival during starvation of EEC-specific *sut1* knockdown animals. **i** Relative whole-body TAG levels of EEC-specific *sut1* knockdown animals. **j** LipidTOX (red) and DAPI (blue) staining of dissected fat body tissue from EEC-specific *sut1* knockdown animals. Scale bar, 50 μm. The number of samples assessed (*n*) is indicated in each graph. For RNAi experiments, *LacZ* knockdown (*TKg*>*LacZ^RNAi*) was used as the negative control. For all bar graphs, mean ± SEM with all data points is shown. For dot blots, the three horizontal lines indicate lower, median, and upper quartiles. Statistics: Wilcoxon rank sum test with Holm's correction (**b**), one-way ANOVA followed by Tukey's multiple comparisons test (**c**), two-tailed Student's *t*-test (**f, g**, and **i**), Log rank test (**h**) *p < 0.05, **p < 0.01, ***p < 0.001; NS, non-significant (p > 0.05). p-values: **b** p < 0.0001 (Fed vs. Starved), p < 0.0001 (Starved vs. Sucrose-refed), p < 0.0001 (Starved vs. Peptone-Refed); **c** p = 0.0472 (Fed vs. Starved), p = 0.9521 (Fed vs. Sucrose-refed), p = 0.2107 (Fed vs. Peptone-refed); **f** p < 0.0001; **g** p = 0.0005; **h** p < 0.0001; **i** p = 0.0136.

results, the fat body-specific RNAi of *AkhR* in *NPF* mutant background improved the sensitivity to starvation and reduction of TAG (Fig. 5i–k). Moreover, double mutant of *AkhR^KO* and *NPF* (*AkhR^KO*; *NPF^sk1/Df*) also improved the reduced lipid phenotype of *NPF* mutants (Fig. 5g). These data indicate that AKH–AKHR signalling is responsible for the metabolic phenotype of animals with loss of NPF or NPFR function.

**NPF/NPFR signalling regulates lipase gene expression in the fat body**. Upon AKH binding, AKHR evokes a rapid and sustained increase in intracellular cAMP and $Ca^{2+}$ accumulation, leading to the activation of multiple lipases that catalyse the hydrolysis of both tri- and diacylglycerides upon starvation[19,20,44]. In the fat body of *D. melanogaster*, two major lipases, Bmm and *Drosophila* hormone sensitive lipase (dHSL), homologues of human adipose triglyceride lipase (ATGL), are involved in regulating TAG amount[19,44,45]. The activities of both Bmm and dHSL are regulated by AKH–AKHR signalling, while their regulatory mechanisms are substantially different[19,20,45–47]. We observed an increase in *Bmm* mRNA expression in loss of *NPF* function animals (Fig. 2f). Consistent with this, *NPFR* knockdown in the CC also increased *Bmm* mRNA expression in the abdomen of females (Fig. 5h). Moreover, co-suppression of *Akh* with *NPFR* in the CC reverted *Bmm* mRNA expression to levels similar to that of the control (Fig. 5h). However, knockdown of *NPFR* in the CC or co-suppression of *NPFR* and *Akh* had no significant effect on *dHSL* mRNA levels (Fig. 5h). Cumulatively, these data suggest that *Bmm*, not *dHSL*, is transcriptionally influenced by NPF/NPFR signalling via AKH.

To assess whether Bmm or dHSL is an effector of activated lipolysis in animals with loss of *NPF* or *NPFR* function, we suppressed *Bmm* or *dHSL* mRNA expression in the fat body cells of *NPF*-null-mutant background. These genetic manipulations were sufficient to rescue the TAG levels of *NPF* mutant animals (Fig. 5i–k). In conjunction with the data showing the NPF-dependent upregulation of *Bmm* mRNA levels, these results suggest that the activity of Bmm is required for NPF/NPFR-regulated lipid mobilisation in the fat body, while dHSL also participates in the regulation of lipid mobilisation, however, in a manner that is independent of NPF/NPFR signalling, at least transcriptionally.

The expression of *Bmm* is reportedly activated by a transcription factor Forkhead box sub-group O (FOXO). FOXO transcriptional activity is tightly associated by its nuclear localisation[46]. Thus, we examined FOXO localisation in fat body cells. Consistent with the increase in *Bmm* mRNA expression,

FOXO nuclear localisation was induced by *TKg*>*NPF^RNAi* or *Akh*>*NPFR^RNAi* (Fig. 6a, b). In contrast, FOXO nuclear localisation in *Akh*>*NPFR^RNAi* was restored with knockdown of *Akh* (Fig. 6b). Moreover, the mRNA level of a FOXO-target gene, *4E-BP* was increased in the abdomen of females, while another FOXO-target gene *Insulin receptor* (*InR*) was not affected (Fig. 6c). These results suggest that NPF-mediated *Bmm* mRNA expression in the fat body may be FOXO-dependent.

**NPF/NPFR signalling control insulin secretion and production**. Since FOXO nuclear localisation is suppressed by insulin signalling pathway[48], the results described above led us to examine the involvement of NPF–NPFR signalling in insulin production and/or secretion. The *D. melanogaster* genome encodes several insulin-like peptide genes (*dilps*). In adulthood, DILP2, DILP3, and DILP5 are produced in and secreted from IPCs in the brain[49,50]. We therefore tested whether NPF from midgut EECs affects DILPs production and secretion. *Dilp3* and *Dilp5* mRNA levels were significantly reduced in *TKg*>*NPF^RNAi* while the level of *Dilp2* mRNA remained constant (Fig. 6d). Since insulin activity is also regulated at the level of DILP secretion[51,52], we assessed accumulation of DILP2, DILP3, and DILP5 in the IPCs with midgut *NPF* knockdown. *NPF* knockdown in the midgut EECs increased DILP2, DILP3, and DILP5 protein levels in the IPCs (Fig. 6e), despite the reduced *Dilp3* and *Dilp5* mRNA levels, indicating that DILPs accumulate in the IPCs. These results suggest that midgut NPF controls *Dilp3* and *Dilp5* mRNA expression, as well as DILPs secretion.

Next, we assessed *NPFR* expression in IPCs. As described above, *NPFR^KI-T2A*-GAL4 and *NPFR^KI-RA/C*-GAL4 are active in many neurons in the brain[37]. We validated *NPFR* expression in the brain in more details and found that both *NPFR^KI-T2A*-GAL4- and *NPFR^KI-RA/C*-GAL4-driven *UAS-GFP* are also expressed in the IPCs (Fig. 7a; Supplementary Fig. 14a). This is consistent with a recent RNA-seq analysis showing that *NPFR* is indeed expressed in the IPCs[53]. We further investigated potential control *Dilps* mRNA expression by NPFR in the IPCs. As expected, *NPFR* knockdown in the IPCs (*Dilp2*>*NPFR^RNAi*), slightly reduced *Dilp2*, *Dilp3* and *Dilp5* mRNA levels, suggesting that midgut NPF controls *Dilps* mRNA expression by directly stimulating the IPCs (Fig. 7b). Similar to *TKg*>*NPF^RNAi* animals, we also confirmed that *NPFR* knockdown in the IPCs (*Dilp2*>*NPFR^RNAi*) induced an accumulation of DILP2 and DILP3 peptide in the IPCs (Fig. 7c). To examine whether DILP2 haemolymph levels are impacted in loss of *NPFR* function animals, we quantified the haemolymph level of circulating endogenous DILP2 tagged with artificial

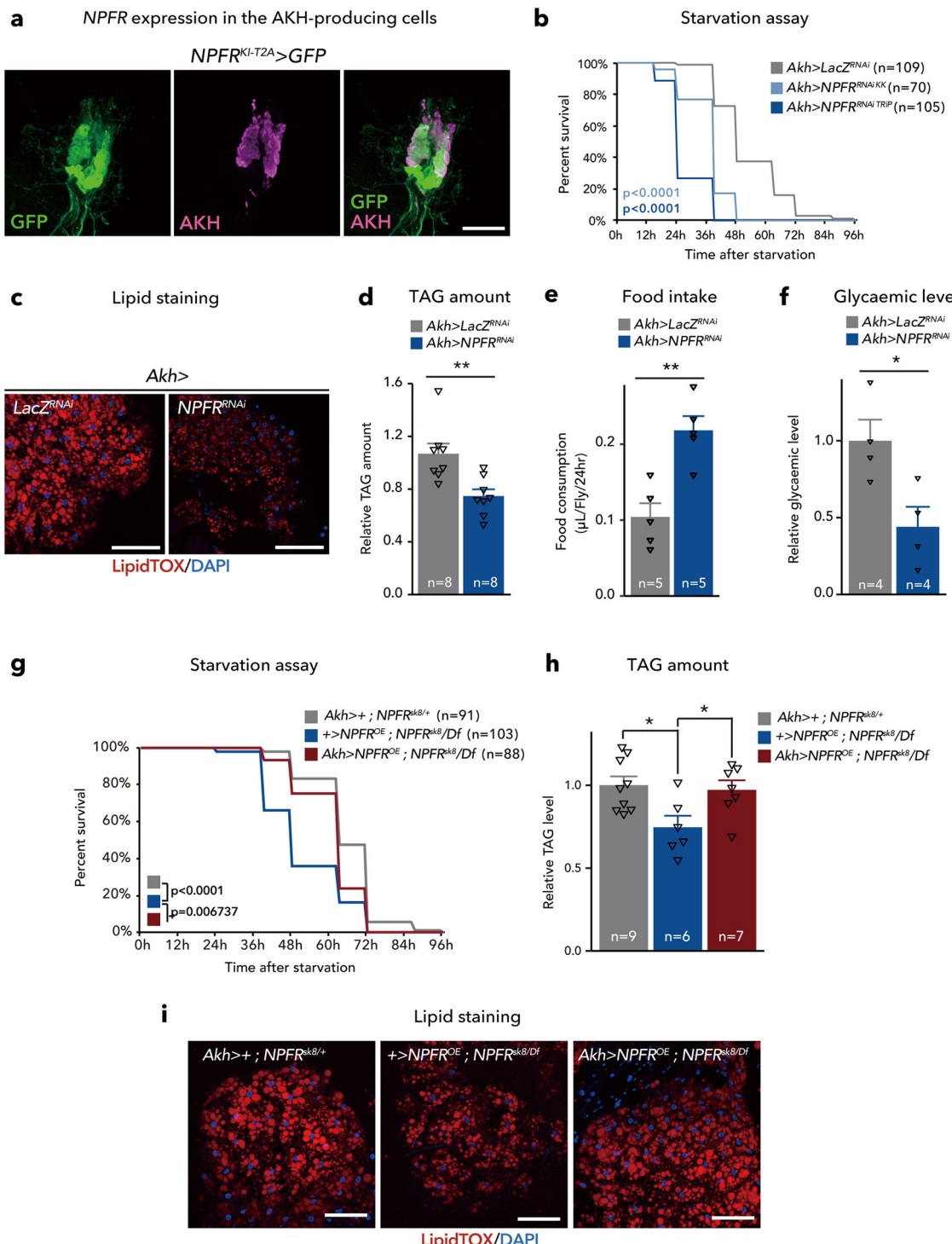

epitopes (DILP2HF)[52,54] in control and *Dilp2>NPFR^RNAi* animals. We observed a significant decrease in circulating DILP2HF in *Dilp2>NPFR^RNAi* animals (Fig. 7d). These results suggest that NPFR in the IPCs positively regulates DILP secretion to the haemolymph.

Since DILP secretion depends on neuronal activities of IPCs[55], we next assessed IPC activity using CaLexA, which allows cumulative tracing of neuronal activity[56], in *ad libitum* fed or starved animals. 24 h starvation significantly attenuated the neuronal activity of IPCs in both control (*Dilp2>CaLexA, LacZ^RNAi*) and *NPFR* knockdown (*Dilp2>CaLexA, NPFR^RNAi*) animals (Fig. 7e). Meanwhile, following ad libitum feeding,

control animals showed robust IPC neuronal activity, whereas knockdown of *NPFR* caused a slight, but significant, reduction in neuronal activity (Fig. 7e). These results demonstrate that NPFR in the IPCs positively regulates DILP secretion by regulating IPCs neuronal activity.

To assess the levels of insulin signalling within peripheral tissue, we used a pleckstrin-homology domain fused to GFP (tGPH), which is recruited to the plasma membrane when insulin signalling is activated[57]. tGPH signal at the plasma membranes of the fat body was significantly reduced in *Dilp2>NPFR^RNAi* animals (Fig. 7f), confirming that DILP secretion is attenuated by *NPFR* knockdown in the IPCs. Consistent with reduced

**Fig. 4 NPFR in the CC is responsible for lipid metabolism. a** Immunofluorescence of corpora cardiaca (CC) in adult flies expressing *UAS-GFP* (green) reporter under *NPFR^KI-T2A-GAL4*. Cell bodies of CC are stained by anti-AKH (magenta). Scale bar, 10 μm. Note, AKH-negative GFP$^+$ cells are the enteric neurons producing sNPF. See Supplementary Fig. 11c. **b** Survival during starvation in flies of control (*Akh>LacZ^RNAi*) and *Akh>NPFR^RNAi*. The number of animals assessed (*n*) is indicated in the graphs. **c** LipidTOX (red) and DAPI (blue) staining of dissected fat body tissue from indicated genotypes. Scale bar, 50 μm. **d** Relative whole-body TAG levels. The number of samples assessed (*n*) is indicated in the graphs. **e** Feeding amount measurement with CAFÉ assay. The number of samples assessed (*n*) is indicated in the graphs. Each sample contained four adult female flies. **f** Relative glycaemic levels in control and *Akh>NPFR^RNAi*. The number of samples assessed (*n*) is indicated in the graphs. **g** Survival during starvation in flies of the indicated genotypes. The number of animals assessed (*n*) is indicated in the graphs. **h** Relative whole-body TAG levels of indicated genotypes. The number of samples assessed (*n*) is indicated in the graphs. **i** LipidTOX (red) and DAPI (blue) staining of dissected fat body tissue from indicated genotypes. Scale bar, 50 μm. For RNAi experiments, *LacZ* knockdown (*Akh>LacZ^RNAi*) was used as negative control. For all bar graphs, mean and SEM with all data points are shown. Statistics: Log rank test with Holm's correction (**b** and **g**), two-tailed Student's *t*-test (**d–f**), one-way ANOVA followed by Tukey's multiple comparisons test (**h**). \**p* < 0.05, \*\**p* < 0.01. *p*-values: **b** *p* < 0.0001 (*Akh>LacZ^RNAi* vs. *Akh>NPFR^RNAiTRiP*), *p* < 0.0001 (*Akh>LacZ^RNAi* vs. *Akh>NPFR^RNAiKK*); **d** *p* = 0.0039; **e** *p* = 0.0024; **f**, *p* = 0.0256; **g**, *p* < 0.0001 (*Akh>+; NPFR^sk8/+* vs. *+>NPFR; NPFR^sk8/NPFR^Df*), *p* < 0.0068 (*+>NPFR; NPFR^sk8/NPFR^Df* vs. *Akh>NPFR; NPFR^sk8/NPFR^Df*); **h** *p* = 0.0183 (*Akh>+; NPFR^sk8/+* vs. *+>NPFR; NPFR^sk8/NPFR^Df*), *p* = 0.0476 (*+>NPFR; NPFR^sk8/NPFR^Df* vs. *Akh>NPFR; NPFR^sk8/NPFR^Df*).

peripheral insulin signalling, *NPFR* knockdown also reduced phospho-AKT levels (Fig. 7g). Together, these data show that NPFR in the IPCs regulates DILP production and secretion, thereby positively controlling the signalling activity of peripheral insulin.

An examination of the effect of *Dilp2>NPFR^RNAi* on metabolism revealed that *NPFR* knockdown in the IPCs caused a mild but significant hypersensitivity to starvation (Fig. 8a). Consistently, TAG level and LipidTOX signal intensity were also reduced in the fat body with *Dilp2>NPFR^RNAi* (Fig. 8b, c). Moreover, *Dilp2>NPFR^RNAi* reduced haemolymph glycaemic level, while feeding amount was significantly increased (Fig. 8d, e). Notably, these metabolic phenotypes of *Dilp2>NPFR^RNAi* were similar to those of *TKg>NPF^RNAi* and *Akh>NPFR^RNAi*. We also confirmed the mRNA expression levels of *Bmm*, *4E-BP*, *InR*, and *pepck1* in the abdomen of *Dilp2>NPFR^RNAi* animals. Despite the reduction of TAG level, *Dilp2>NPFR^RNAi* failed to increase *Bmm* mRNA expression (Fig. 8f), suggesting that the lean phenotype of *Dilp2>NPFR^RNAi* animal is not due to an increase in *Bmm* mRNA expression. However, expression of other FOXO-target genes, *4E-BP* and *pepck1* were upregulated with *Dilp2>NPFR^RNAi* (Fig. 8f). Consistent with this, *Dilp2>NPFR^RNAi* induced FOXO nuclear localisation (Fig. 8g). These data suggest that NPFR in the IPCs regulates DILPs expression and secretion, followed by nuclear translocation of FOXO in the fat body to alter some FOXO-target genes.

Since IPCs produce multiple neuropeptides, including DILPs and Drosulfakinin (Dsk), we next sought to identify which neuropeptide in the IPCs is responsible for NPF/NPFR-mediated regulation of lipid storage in the fat body. Results show that knockdown of *dilp3* (*Dilp2>dilp3^RNAi*) resulted in significant reduction of TAG abundance, while the others had no significant effect (Supplementary Fig. 14b). Our data is consistent with a previous study demonstrating that *dilp3* mutant animals exhibit reduced TAG levels[58]. Additionally, although Dsk is known to regulate feeding behaviour in adults[59], *dsk* expression was not affected by *NPFR* knockdown in IPCs (Supplementary Fig. 14c).

Our data indicates that *NPFR* knockdown in the CC resulted in a stronger hypersensitive phenotype to starvation compared to that detected following *NPFR* knockdown in the IPCs (Figs. 4b and 8a). To explain this discrepancy, we hypothesised that *NPFR* knockdown in the CC might lead to a significant alteration in DILP production within IPCs. To test this hypothesis, we quantified *dilps* mRNA levels in *Akh>NPFR^RNAi* and found that *NPFR* knockdown in the CC decreased *dilp3* and *dilp5* mRNA levels (Supplementary Fig. 14d). In contrast, *NPFR* knockdown in the IPCs (*Dilp2>NPFR^RNAi*) did not influence *Akh* mRNA expression (Supplementary Fig. 14e). Together, these data suggest

that *NPFR* knockdown in the CC results in not only enhanced AKH production, but also suppression of DILP production.

**NPF neurons might not play a crucial role in AKH and DILPs production**. Although *NPF* knockdown in the brain did not exhibit significant effects in metabolism Supplementary Fig. 3), it remains possible that brain NPF participates in the regulation of AKH and DILPs. However, three lines of evidence as follows are likely to negate this possibility. First, we confirmed AKH and DILP mRNA and protein levels following brain-specific *NPF* knockdown (*fbp>NPF^RNAi*). Consistent with the metabolic phenotype, *NPF* knockdown in the brain did not impact mRNA or protein levels of either AKH or DILPs (Supplementary Fig. 15a–d). Second, postsynaptic trans-Tango signals driven by *NPF-GAL4* were not detected in CC cells or neurons in the PI region (Supplementary Fig. 15e, f). Third, 24 h starvation did not affect NPF protein levels in the brain (Supplementary Fig. 15g). Taken together, these data suggest that brain NPF neurons do not affect AKH and DILPs levels.

Taken together, our findings suggest that midgut-derived, but not neuronal NPF, binds NPFR in the CC and IPCs, suppressing AKH production and enhancing DILP secretion, respectively. As a result, midgut NPF employs downstream FOXO-target genes to regulate carbohydrate and lipid metabolism through glucagon and insulin, respectively (Fig. 9).

## Discussion

Here, we demonstrated that midgut-derived NPF acts as a sensor of dietary sugar and plays an important role in the regulation of adult carbohydrate and lipid homoeostasis in *D. melanogaster*. Importantly, we showed that midgut NPF is received by the CC and IPCs, to coordinate their expression of glucagon-like and insulin-like hormones, respectively. Previous studies reported that midgut EEC-derived Activin-β and Bursα are important for carbohydrate and lipid metabolism in *D. melanogaster*, although these enteroendocrine hormones have not been shown to directly act on the CC or IPCs. Activin-β acts on the fat body to regulate *AkhR* expression in the larval fat body[9]. Bursα is secreted in response to dietary sugars, but it is received by un-characterised neurons that express its receptor, *Lgr2*, leading to suppression of *Akh* expression[11]. We therefore propose that NPF is the first incretin-like hormone in invertebrates, and its production and secretion are stimulated by dietary nutrients similar to incretins (Fig. 9).

**Nutrient-dependent NPF regulation**. Due to technical limitations, we were unable to quantify the haemolymph titre of NPF

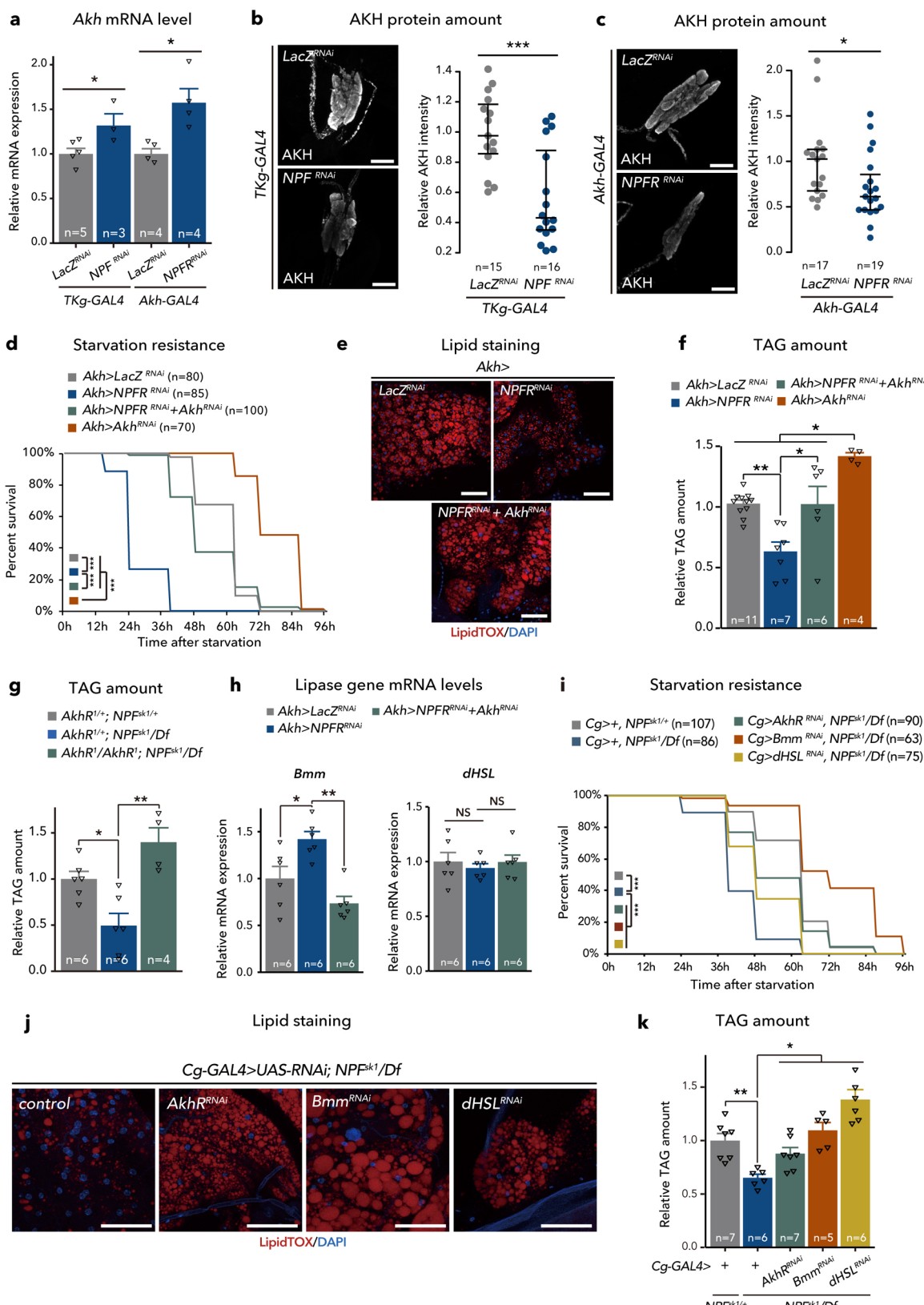

and, therefore, did not examine whether midgut NPF contributes to the NPF haemolymph level. Nevertheless, our data strongly suggests that dietary sugar controls not only midgut *NPF* expression but also NPF secretion from the midgut. In this scenario, NPF secretion is attenuated in starved conditions, while the attenuation is restored by sugar re-feeding.

We found that Sut1, a homologue of mammalian SLC2, is a regulator of sugar-dependent NPF production in EECs. Considering that Sut1 is localised on plasma membranes and contributes to the elevation of intracellular glucose levels, it is likely that Sut1 transports glucose into the cell. Similar to Sut1, Glut1, another mammalian SLC2 homologue, acts as a glucose

**Fig. 5 NPF/NPFR signalling regulates metabolic homoeostasis through Akh/AkhR signalling. a** RT-qPCR analysis of *Akh* mRNA expression following *TKg-GAL4*-mediated knockdown of *NPF* or *Akh-GAL4*-mediated knockdown of *NPFR*. The number of samples assessed (*n*) is indicated in the graphs. **b, c** Immunostaining and quantification of AKH protein level (white) in adult CC of *TKg-GAL4*-mediated knockdown of *NPF* (*TKg>NPF^RNAi*) (**b**) or *Akh-GAL4*-mediated knockdown of *NPFR* (*Akh>NPFR^RNAi*) (**c**). The number (*n*) of CCs analysed in each genotype is indicated in the graph. Scale bar, 20 µm. **d, i** Survival during starvation in flies of each genotype. The number of animals assessed (*n*) is indicated in the graphs. **e, j** LipidTOX (red) and DAPI (blue) staining of dissected fat body tissue from indicated genotypes. Scale bar, 50 µm in **c**, 20 µm in **j**. **f, g, k**, Relative whole-body TAG levels of each genotype. The number of animals assessed (*n*) is indicated in the graphs. **h** RT-qPCR analysis of *Bmm* (left) and *dHSL* (right) mRNA levels in the abdomens dissected from each genotype. The number of samples assessed (*n*) is indicated in the graphs. For RNAi experiments, *LacZ* knockdown (*TKg>LacZ^RNAi* and *Akh>LacZ^RNAi*) was used as negative control. For all bar graphs, mean and SEM with all data points are shown. For dot blots, the three horizontal lines indicate lower, median, and upper quartiles. Statistics: two-tailed Student's *t*-test (**a-c**), Log rank test with Holm's correction (**d** and **i**), one-way ANOVA followed by Tukey's multiple comparisons test (**f, g, h**, and **k**). *$p < 0.05$, **$p < 0.01$, ***$p < 0.001$; NS, non-significant ($p > 0.05$). *p*-values: **a** $p = 0.0470$ (*TKg>LacZ^RNAi* vs. *TKg>NPF^RNAi*), $p = 0.0142$ (*Akh>LacZ^RNAi* vs. *Akh>NPFR^RNAi*); **b** $p = 0.0007$; **c** $p = 0.0300$; **d** $p < 0.0001$ (*Akh>LacZ^RNAi* vs. *Akh>NPFR^RNAi*), $p < 0.0001$ (*Akh>NPFR^RNAi* vs. *Akh>NPFR^RNAi + Akh^RNAi*), $p < 0.0001$ (*Akh>LacZ^RNAi* vs. *Akh>Akh^RNAi*, *Akh>NPFR^RNAi* vs. *Akh>Akh^RNAi*, *Akh>NPFR^RNAi + Akh^RNAi* vs. *Akh>Akh^RNAi*); **f** $p = 0.0028$ (*Akh>LacZ^RNAi* vs. *Akh>NPFR^RNAi*), $p = 0.0108$ (*Akh>NPFR^RNAi* vs. *Akh>NPFR^RNAi + Akh^RNAi*), $p = 0.0155$ (*Akh>LacZ^RNAi* vs. *Akh>Akh^RNAi*), $p < 0.0001$ (*Akh>NPFR^RNAi* vs. *Akh>Akh^RNAi*), $p = 0.0294$ (*Akh>NPFR^RNAi + Akh^RNAi* vs. *Akh>Akh^RNAi*); **g** $p = 0.0208$ (*AkhR^1/+; NPF^sk1/+* vs. *AkhR^1/+; NPF^sk1/NPF^Df*), $p = 0.0007$ (*AkhR^1/+; NPF^sk1/NPF^Df* vs. *AkhR^1/ AkhR^1; NPF^sk1/NPF^Df*); **h** *Bmm*, $p = 0.0218$ (*Akh>LacZ^RNAi* vs. *Akh>NPFR^RNAi*), $p = 0.0005$ (*Akh>NPFR^RNAi* vs. *Akh>NPFR^RNAi + Akh^RNAi*), *dHSL*, $p = 0.7966$ (*Akh>LacZ^RNAi* vs. *Akh>NPFR^RNAi*), $p = 0.8188$ (*Akh>NPFR^RNAi* vs. *Akh>NPFR^RNAi+Akh^RNAi*); **i** $p < 0.0001$ (*Cg>+; NPF^sk1/+* vs. *Cg>+; NPF^sk1/NPF^Df*), $p < 0.0001$ (*Cg>+; NPF^sk1/NPF^Df* vs. *Cg>AkhR^RNAi; NPF^sk1/NPF^Df*), $p < 0.0001$ (*Cg>+; NPF^sk1/NPF^Df* vs. *Cg>Bmm^RNAi; NPF^sk1/NPF^Df*), $p < 0.0001$ (*Cg>+; NPF^sk1/NPF^Df* vs. *Cg>dHSL^RNAi; NPF^sk1/NPF^Df*); **k** $p = 0.0073$ (*Cg>+; NPF^sk1/+* vs. *Cg>+; NPF^sk1/NPF^Df*), $p = 0.0230$ (*Cg>+; NPF^sk1/ NPF^Df* vs. *Cg>AkhR^RNAi; NPF^sk1/NPF^Df*), $p = 0.0015$ (*Cg>+; NPF^sk1/ NPF^Df* vs. *Cg>Bmm^RNAi; NPF^sk1/NPF^Df*), $p < 0.0001$ (*Cg>+; NPF^sk1/ NPF^Df* vs. *Cg>dHSL^RNAi; NPF^sk1/NPF^Df*).

transporter to elevate intracellular glucose levels in *D. melanogaster*[35]. Additionally, Glut1 has been shown to be essential for nutrient-dependent production and secretion of Bursα from EECs[11]. However, our data suggest that Glut1 does not affect NPF production (Supplementary Fig. 6b, c). Importantly, NPF and Bursα are produced in different regions of the midgut, namely in the anterior and posterior midgut, respectively[11,60]. Therefore, different subtypes of EECs appear to have different glucose sensing systems. Thus, characterising how the differences in EEC sugar sensing systems affect the metabolic robustness of individuals may clarify the significance of the more than 30 SLC2 genes in *D. melanogaster*[29].

In mammalian EECs, especially GLP-1+ L cells, dietary glucose is transported by glucose transporter, Glut2, and sodium coupled glucose transporter 1 (SGLT-1) to stimulate GLP-1 secretion[27,28,61,62]. In addition to sugars, fatty acids and amino acids also stimulate GLP-1 secretion from mammalian EECs[28,63,64]. Therefore, *D. melanogaster* EECs might also be regulated by multiple regulatory systems in response to different nutrient types. However, these systems remain largely undefined. For example, in this study, we were unable to determine the underlying mechanism by which midgut NPF mRNA and protein levels are significantly upregulated by peptone feeding (Fig. 3a–c). Future studies should offer a more comprehensive investigation of nutrient-dependent enteroendocrine hormone regulation at the molecular level.

**Metabolic function of NPF/NPFR.** Our data demonstrated that midgut-derived NPF-controlled organismal carbohydrate and lipid metabolism through AKH and insulin signalling. Animals with loss of *NPF* function were in a catabolic state, reminiscent of starved animals, as judged by the following observation from our RNA-seq and metabolome analyses: (1) upregulation of glycolysis, TCA cycle, mitochondrial respiratory chain complex genes, and starvation-induced genes, (2) increase of several TCA cycle metabolites, (3) lipodystrophy and hypoglycaemia along with hyperphagia, and (4) nuclear localisation of FOXO and the induction of starvation-induced FOXO-target genes. These phenotypes are likely due to upregulation of AKH/AKHR signalling and attenuation of insulin signalling in the peripheral tissues. Taken together, our results suggest that NPFR in the CC and IPCs

has pivotal role in the regulation of organismal TAG and glycaemic levels.

In the adult fat body, TAG level is controlled by two lipases, dHSL and Bmm in a redundant manner. Given that knockdown of either *Bmm* or *dHSL* in the fat body restored TAG reduction in *NPF*-null-mutant animals to the control level, we hypothesise that both lipases cooperatively control lipid breakdown in the NPF–NPFR axis. These data support our idea that glucose and stored lipids are mobilised to the TCA cycle to generate energy in animals with loss of *NPF* function.

**Cross talk with other signalling.** One of our striking findings is that NPF produced by midgut EECs directly stimulated the CC and IPCs, indicating the presence of both a midgut–CC–fat body axis and a midgut–IPCs–fat body axis in *D. melanogaster*. Although many studies have demonstrated that neuronal signalling in the brain and humoral factors from peripheral tissues stimulate either CC or IPCs[14,65], factors that stimulate both the CC and IPCs are less defined. As the one and only example of such factors, it was recently reported that sNPF from two pairs of neurons directly innervating both the CC and IPCs controls glycaemic level in a sugar-responsive manner[43]. sNPF receptor (sNPFR) is expressed and coupled with a trimeric G protein signalling in the CC and IPCs, leading to the suppression of AKH secretion and enhancement of DILP2 secretion. Although NPFR is coupled with Gαq and Gαi subunits in heterologous expression systems[66,67], it remains unclear which trimeric G protein is coupled in the CC and IPCs to transmit NPF signals. Further studies are needed to investigate the integration of neuron-derived sNPF and midgut-derived NPF to adequately stimulate the CC and IPCs for the regulation of AKH and DILP secretion, respectively.

Many studies have identified and characterised factors that regulate DILP production and secretion. Dietary nutrients, neuropeptide signalling, and adipocyte-derived factors regulate IPCs to coordinate systemic growth and energy-related events[14,50,65,68]. In contrast, much less is known about factors regulating AKH production and secretion. In the process of peer review of this paper, a preprint manuscript reported that Allatostatin C (AstC) from EECs regulates AKH production or secretion[69]. Beside the neuropeptide signalling described above (NPF, sNPF, Bursα, AstC), other signalling components also

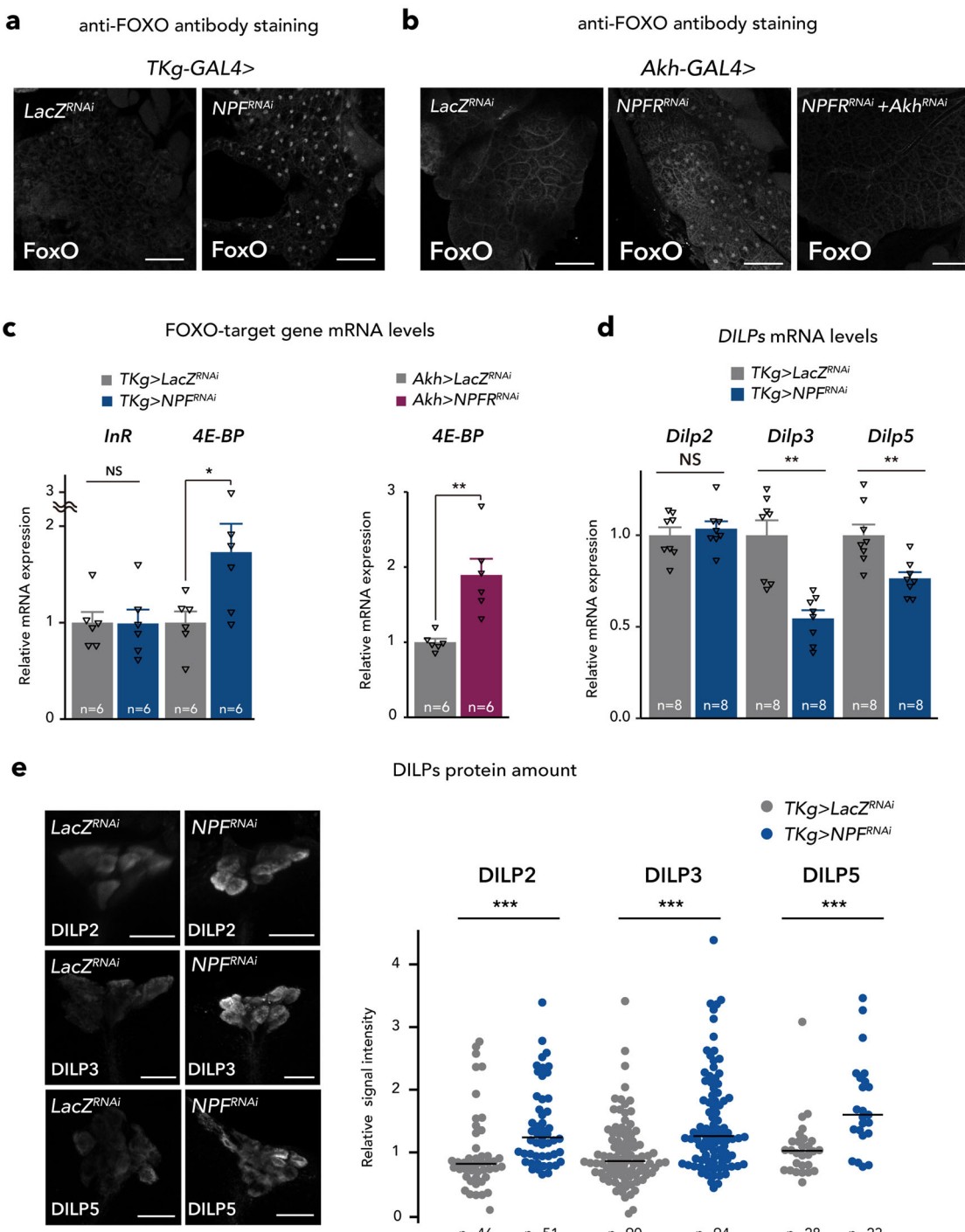

**Fig. 6 Midgut-derived NPF controls DILPs level. a**, **b** FOXO (white) immunostaining of the fat body in adult flies of each genotype. Scale bar, 20 μm. Note that FOXO nuclear localisation was induced in *TKg>NPF^RNAi^* and *Akh>NPFR^RNAi^*. **c** RT-qPCR analysis of FOXO-target gene mRNA levels in the abdomens dissected from each genotype. The number of samples assessed (*n*) is indicated in the graphs. **d** RT-qPCR analysis of *Dilps* mRNA level following *TKg-GAL4* mediated knockdown of *NPF*. The number of samples assessed (*n*) is indicated in the graphs. **e** DILP2, 3, and 5 (white) immunostaining and quantification in the brain of adult flies of *TKg-GAL4*-mediated *NPF* RNAi animals. Scale bar, 20 μm. The number of samples assessed (*n*) is indicated in the graphs. For RNAi experiments, *LacZ* knockdown (*TKg>LacZ^RNAi^* and *Akh>LacZ^RNAi^*) was used as negative control. For all bar graphs, mean and SEM with all data points are shown. For dot blots, the horizontal lines indicate median quartile. Statistics: two-tailed Student's *t*-test (**c**–**e**), Wilcoxon rank sum test (**f**). *$p < 0.05$, **$p < 0.01$, ***$p < 0.001$; NS, non-significant ($p > 0.05$). *p*-values: **c** (left), $p = 0.9604$ (*InR*), $p = 0.0437$ (*4E-BP*); **c** (right), $p = 0.0023$ (*4E-BP*); **d** $p = 0.5609$ (*Dilp2*), $p = 0.0003$ (*Dilp3*), $p = 0.0036$ (*Dilp5*); **e** $p < 0.0001$ (DILP2), $p < 0.0001$ (DILP3), $p < 0.0001$ (DILP5).

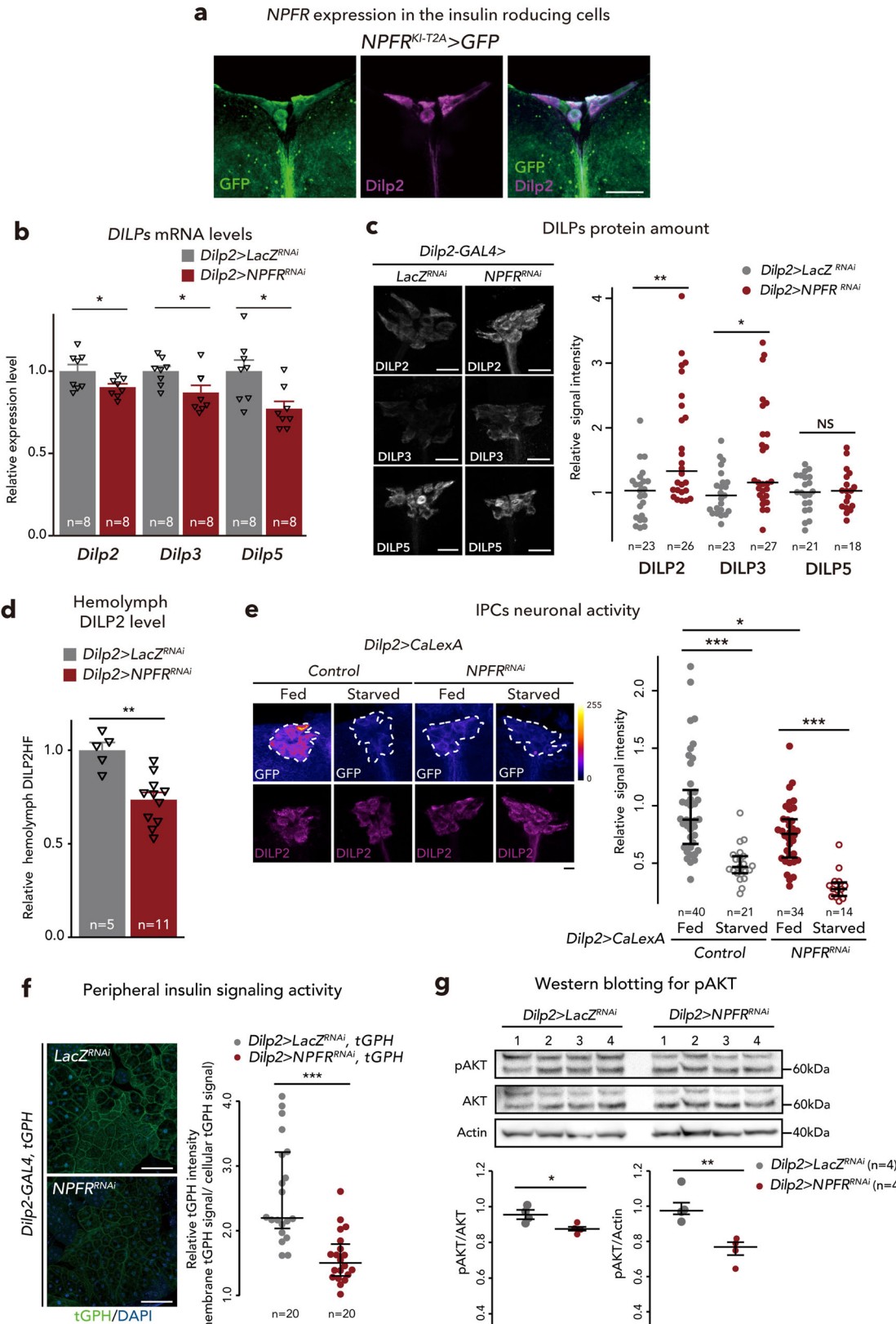

**a** *NPFR expression in the insulin roducing cells*

$NPFR^{KI-T2A}>GFP$

**b** *DILPs mRNA levels*

**c** DILPs protein amount

**d** Hemolymph DILP2 level

**e** IPCs neuronal activity

**f** Peripheral insulin signaling activity

**g** Western blotting for pAKT

function in the CC, such as Allatostatin A-receptor 2, a water sensor encoded by *pickpocket28*, and a $H_2O_2-$ and ultraviolet light-sensitive isoform of *TrpA1*[70–72]. Moreover, AKH secretion is regulated by a neurotransmitter secreted from *Lgr2*[+] neurons, although the neurotransmitter has not been characterised[11]. Studies that detail the signalling cross talk between the factors that control AKH actions are required, to further elucidate how CC cells sense multiple nutritional and physiological cues to control carbohydrate and lipid metabolism. In fact, CC reportedly produces another peptide, Limostatin (Lst), that stimulates IPCs to suppress production of DILPs, including decretin[73]. Therefore, Lst may participate in the downregulation of *dilp3* and *dilp5*

**Fig. 7 NPFR in the insulin-producing cells regulates DILPs level. a** Immunofluorescence of the IPCs in adult flies expressing *UAS-GFP* (green) reporter under *NPFR^{KI-T2A}-GAL4*. Cell bodies of IPCs are stained by anti-DILP2 (magenta). Scale bar, 20 µm. **b** RT-qPCR analysis of *Dilps* mRNA level following *Dilp2-GAL4*-mediated knockdown of *NPFR*. The number of samples assessed (*n*) is indicated in the graphs. **c** DILP2, 3, and 5 (white) immunostaining in the brain of adult flies of *Dilp2-GAL4*-mediated *NPFR* RNAi animals. Scale bar, 20 µm. The sample number (*n*) analysed in each genotype is indicated in the graph. **d** Measurements of circulating DILP2HF abundance. The number of samples assessed (*n*) is indicated in each graph. **e** (left) Immunofluorescent staining of IPCs following *Dilp2-GAL4*-mediated *NPFR* knockdown, including overexpression of the Ca$^{2+}$ sensor CaLexA. Fluorescence signals are pseudocoloured; high (Max: 255) to low (Minimum: 0) intensity is displayed as warm (yellow) to cold (blue) colours with a colour scale. IPCs were visualised by immunostaining with anti-DILP2 antibody (magenta). IPCs are marked by white dashed line. Scale bar, 20 µm. (right) Quantification of the CaLexA signal intensity normalised by ad libitum feeding controls. The sample number (*n*) analysed in each genotype is indicated in the graph. **f** (left) Immunofluorescence staining in the fat bodies of adults expressing the insulin signalling sensor tGPH (green) following *Dilp2-GAL4*-mediated *NPFR* knockdown. Scale bar, 50 µm. (right) Quantification of tGPH levels. The relative tGPH level is defined as membrane tGPH intensity divided by cellular tGPH intensity. Each point represents signal intensity of a single fat body cell. The number (*n*) analysed in each genotype is indicated in the graph. **g** Western blotting analysis of phospho-AKT, pan-AKT, and Actin. The expected protein size of non-phosphorylated AKT is 59.92 kDa. The sample number (*n*) analysed in each genotype is indicated in the graph. Full scan images of blot are represented in the Source Data file. For RNAi experiments, *LacZ* knockdown (*Dilp2>LacZ^{RNAi}*) was used as negative control. For all bar graphs, mean and SEM with all data points are shown. Statistics: two-tailed Student's *t*-test (**b**, **d**, and **g**), Wilcoxon rank sum test with Holm's correction (**c**, **e**, and **f**). *$p < 0.05$, **$p < 0.01$, ***$p < 0.001$; NS, non-significant ($p > 0.05$). *p*-values: **b** $p = 0.0452$ (*Dilp2*), $p = 0.0264$ (*Dilp3*), $p = 0.0132$ (*Dilp5*); **c** $p = 0.0028$ (DILP2), $p = 0.0118$ (DILP3), $p = 0.8783$ (DILP5); **d** $p = 0.0012$; **e** $p < 0.0001$ (Control Fed vs. Control Starved), $p = 0.0099$ (Control Fed vs. *NPFR^{RNAi}* Fed), $p < 0.0001$ (*NPFR^{RNAi}* Fed vs. *NPFR^{RNAi}* Starved); **f** $p < 0.0001$, **g** (left) $p = 0.0222$, (right) $p = 0.0067$.

expression following *NPFR* knockdown in the CCs (Supplementary Fig. 13b).

**Midgut NPF vs. brain NPF.** Our previous study[17], as well as the results of the current study, confirm the significant biological function of midgut NPF in *D. melanogaster*. Meanwhile, many previous studies have reported that brain NPF has versatile roles in the feeding and social behaviour of insects[23,74,75]. Therefore, the metabolic phenotypes of *NPF* genetic mutants may reflect the diverse functions of brain NPF, although the data from the current study does not support this postulate (Supplementary Figs. 3 and 15). In particular, brain-specific *NPF* knockdown does not phenocopy *NPF* mutation or midgut-specific *NPF* knockdown. These data imply distinct physiological functions between midgut and brain NPF.

Another key finding in this study is the anorexigenic function of midgut-derived NPF, which is in contrast to the orexigenic function of brain NPF[22,23]. Interestingly, agonists of NPY-like receptor 7 disrupt host-seeking behaviour and biting in the yellow fever mosquito, *Aedes aegypti*[76]. Moreover, disruption of NPF/NPFR signalling results in abnormal feeding behaviour and reduced growth in several insects[75,77,78]. Since other insects also produce NPF from the brain and gut[77,79], it is important to validate the source of circulating NPF and discriminate the function of brain NPF from that of gut-derived NPF.

**Commonality with mammalian system.** A growing number of evidences have demonstrated that, similar to mammals, the *D. melanogaster* intestine plays versatile roles in systemic physiology[80]. Although it is simpler than the mammalian gastrointestinal tract, the *D. melanogaster* intestinal epithelium is functionally regionalised and displays similarity both at the cellular and molecular levels[30,31,81]. In mammals, GIP from K-cells (largely in the upper small intestine) and GLP-1 from L cells (predominantly in the distal small and large intestine) are considered incretins, which induce insulin secretion by stimulating β cells in the pancreatic islets[5,6,61]. Among incretins, GLP-1 suppresses glucose-dependent glucagon secretion via its receptor GLP-1R in α-cells of the pancreas[82]. Although *D. melanogaster* endocrine system is different from that of mammals, we propose that midgut-derived NPF have similar role in insulin/glucagon regulation as mammalian GLP-1. Treatment with GLP-1 agonists reduces food intake and hunger, promoting fullness and satiety

with the ultimate result of weight loss in patients with obesity or type 2 diabetes[13,83,84]. Similar to this, gut-derived NPF regulated satiety in *D. melanogaster* in our study. However, loss of GLP-1/GLP-1R signalling has a non-significant effect on weight and fat mass in regular food-fed mice, whereas loss of NPF/NPFR resulted in lean phenotype in regular food. Thus, although there are substantial similarities in the physiological function of mammalian incretins and *D. melanogaster* NPF, their effects on metabolism are divergent in some aspects. Considering that GLP-1 acts on many organs and tissues, including the nervous system, heart, stomach, gut, and pancreas[5], and that *NPFR* is expressed in the nervous system, visceral muscles, and EECs of the gut[31], differences in the inter-organ communication systems of mammals and *D. melanogaster* in the GLP-1 and NPF may produce differences in the physiological effects of these enteroendocrine hormones. To further understand midgut-derived NPF-dependent inter-organ communication system, it would be intriguing to investigate the role of NPFR in potential target tissues, such as visceral muscles of the gut and *NPFR*$^+$ neurons, other than the IPCs. The ease of tissue-specific genetic manipulations, together with the evolutionary conservation of central signalling pathways regulating metabolism and energy homoeostasis, makes *D. melanogaster* a powerful model system to unravel the role of incretin-like enteroendocrine hormones in systemic organismal metabolism.

## Methods

**Fly stock and husbandry.** Flies were raised on a fly food (5.5 g agar, 100 g glucose, 40 g dry yeast, 90 g cornflour, 3 mL propionic acid, and 3.5 mL 10% butyl *p*-hydroxybenzoate (in 70% ethanol) per litre) in a 12/12 h light/dark condition at 25 °C for 6 days before experiments. Virgin female flies were used for all fly experiments.

The following transgenic and mutant stocks were used: *NPF^{sk1}* and *NPFR^{sk817}*), *NPF^{Df(3R)ED10642}* (Kyoto stock center [DGRC] #150266), *NPFR^{Df(3R)BSC464}* (Bloomington stock center [BDSC] #24968), *Akh^{KO}* (a gift from Yi Rao, Peking University School of Life Sciences, China)[37], *Akh^{A85}*, *AkhR^1* (gifts from Ronald P. Kühnlein, Max-Planck-Institut für Biophysikalische Chemie, Germany)[20], *sut1^{KO}* (this study), *tub>FRT>GAL80>FRT* (BDSC# 38879), *Otd-FLP* (a gift from Daisuke Yamamoto, National Institute of Information and Communications Technology, Japan)[86], *Tk-gut-GAL4*[18], *UAS-LacZ^{RNAi}*[87] (gifts from Masayuki Miura, the University of Tokyo, Japan), *nSyb-GAL4* (BDSC#51941), *Akh-GAL4* (BDSC#25683), *dilp2-GAL4* (BDSC#37516), *how-GAL4* (BDSC#1767), *tj-GAL4* (Kyoto stock center #104055), *NPFR^{KI-T2A}-GAL4*[33], *NPFR^{KI-RA/RC}-GAL4*[37] (BDSC#84672), *fbp-GAL4*[24] (a gift from Chika Miyamoto and Hubert Amrein, Texas A&M University, USA), *sut1^{KI-T2A}-GAL4* (this study), *tub-GAL80^{ts}* (BDSC#7019) *UAS-NPF*, *UAS-NPFR* (a gift from Ping Shen, University of Georgia, USA), *UAS-mCD8::GFP* (BDSC#32186), *UAS-FLII12Pglu-700µδ6* (a gift from

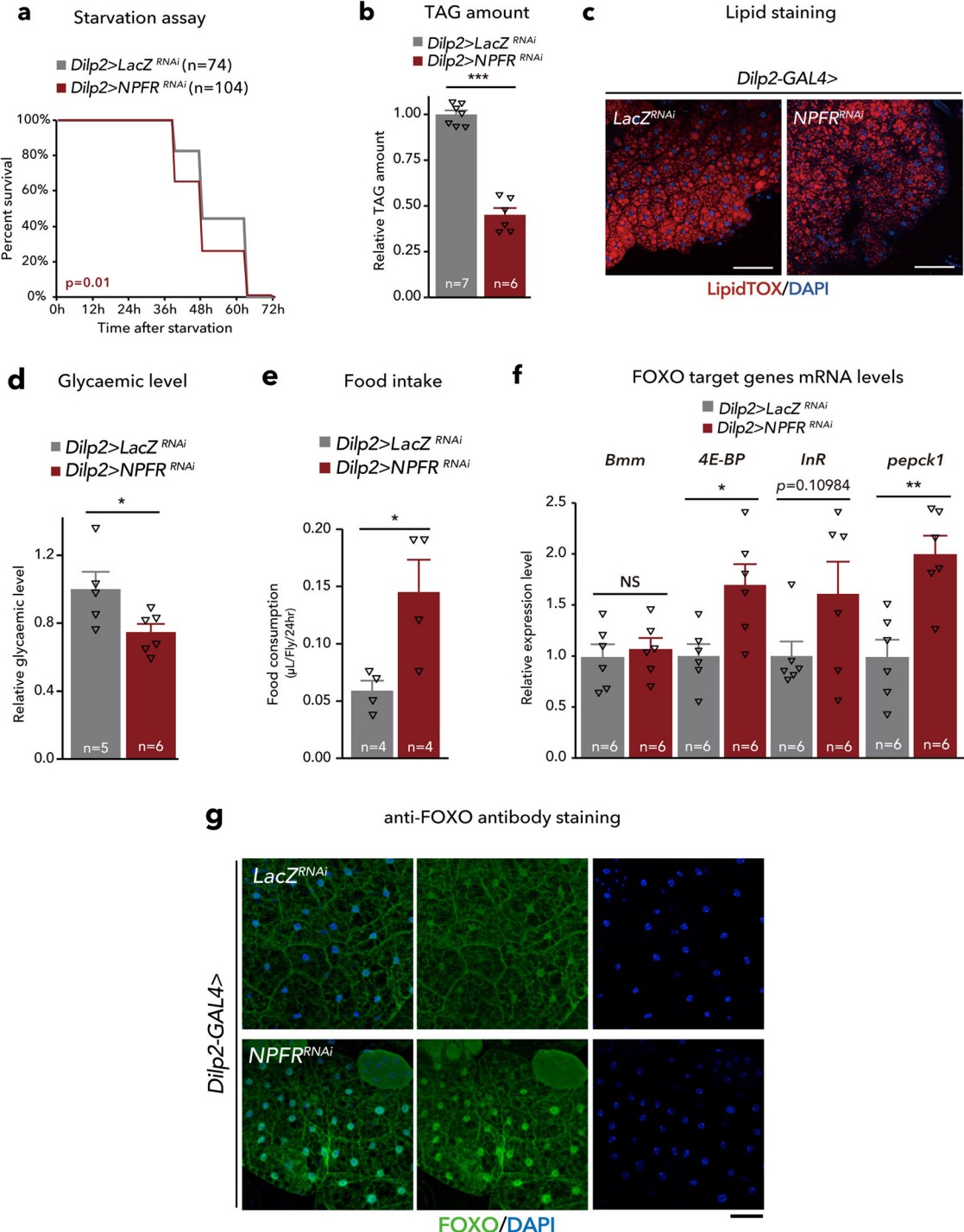

**Fig. 8 NPFR in the insulin-producing cells regulates carbohydrate/lipid metabolism. a** Survival during starvation in flies of each genotype. The number of animals assessed (*n*) is indicated in the graphs. **b** Relative whole-body TAG levels of each genotype. The number of animals assessed (*n*) is indicated in the graphs. **c** LipidTOX (red) and DAPI (blue) staining of dissected fat body tissue from indicated genotypes. Scale bar, 20 μm. **d** Relative glycaemic levels in control and *Dilp2>NPFR^RNAi^*. The number of samples assessed (*n*) is indicated in the graphs. **e** Feeding amount measurement of each genotype with CAFÉ assay. *n* = 4 samples, each point contained four adult female flies. **f** RT-qPCR analysis of FOXO-target gene mRNA levels in the fat body dissected from each genotype. The number of samples assessed (*n*) is indicated in the graphs. **g** FOXO (white) immunostaining of the fat body in adult flies of each genotype. Scale bar, 50 μm. For RNAi experiments, *LacZ* knockdown (*Dilp2>LacZ^RNAi^*) was used as negative control. For all bar graphs, mean and SEM with all data points are shown. For dot blots, the three horizontal lines indicate lower, median, and upper quartiles. Statistics: Log rank test (**a**), two-tailed Student's *t*-test (**b**, **d**–**f**). *\*p* < 0.05, \*\**p* < 0.01, \*\*\**p* < 0.001; NS, non-significant (*p* > 0.05). *p*-values: **a**, *p* = 0.0100; **b** *p* < 0.0001; **d** *p* = 0.0417; **e** *p* = 0.0269; **f** *p* = 0.6468 (*Bmm*), *p* = 0.0146 (*Thor*), *p* = 0.1098 (*InR*), *p* = 0.0024 (*pepck1*).

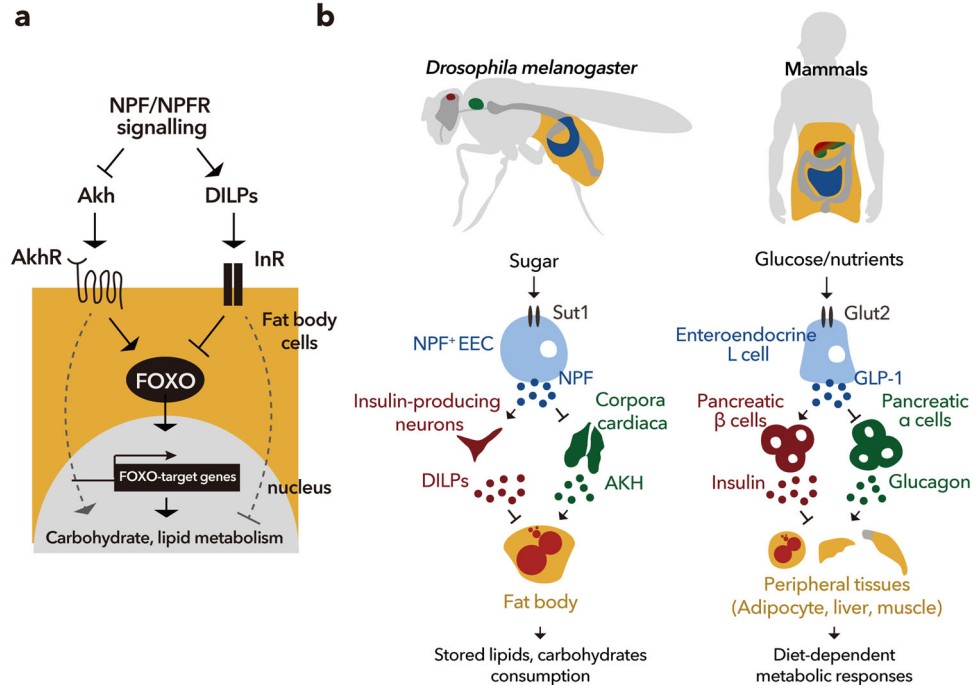

**Fig. 9 Midgut-derived NPF regulates AKH and DILPs level in response to dietary sugar. a** A working model illustrating dual pathway coordination of FOXO and FOXO-target genes. (i) The loss of NPF/NPFR signalling in the CC—enhanced AKH/AKHR signalling induces FOXO nuclear localisation and carbohydrate/lipid metabolism—and (ii) the loss of NPF/NPFR signalling in the IPCs—attenuation of insulin signalling induces FOXO nuclear localisation and enhances carbohydrate/lipid metabolism. The balance of AKH/AKHR and insulin signalling is coordinated by gut-derived NPF that responds dietary sugars. **b** (left) In *D. melanogaster*, enteroendocrine cells respond to the dietary sugars by secreting a neuropeptide, NPF, which signals via its neuronal receptor NPFR. NPF/NPFR signalling regulates energy consumption through dual neuronal relay which are restriction of glucagon-like, AKH production, and enhancement of insulin-like peptides (DILPs). Subsequent modulation of AKHR and insulin signalling within the fat body/adipose tissue maintains lipid/carbohydrate catabolism; thus, impaired NPF/NPFR signalling leads to depletion of energy stores. (right) The EEC-IPC and the EEC-CC axes in *D. melanogaster* are similar to the gut EECs-pancreas axis in mammals. Mammalian enteroendocrine hormone, GLP-1 also controls insulin and glucagon levels in response to dietary nutrients.

Chika Miyamoto and Hubert Amrein)[35], *UAS-trans-Tango* (BDSC#77480), *UAS-sut1::mVenus* (this study), *CaLexA* (BDSC#66542), and *tGPH* (BDSC#8164). RNAi constructs targeting *UAS-NPF*[RNAiKK] (VDRC#108772), *UAS-NPF*[RNAiTRiP] (BDSC#27237), *UAS-NPFR*[RNAiKK] (VDRC#107663), *UAS-NPFR*[RNAiTRiP] (BDSC#25939), *UAS-sut1*[RNAi] (VDRC#104983), *UAS-sut1*[RNAiTRiP] (BDSC#65964), *UAS-Glut1*[RNAi] (VDRC#101365), *UAS-Akh*[RNAi] (VDRC#105063), *UAS-AkhR*[RNAi] (VDRC#109300), *UAS-Bmm*[RNAi] (VDRC#37877), *UAS-dHSL*[RNAi] (VDRC#109336), *UAS-dilp2*[RNAi] (VDRC#102158), *UAS-dilp3*[RNAi] (VDRC#106512), *UAS-dilp5*[RNAi] (VDRC#105004), and *UAS-dsk*[RNAi] (VDRC#106592).

Except in Fig. 1a, *TKg-GAL4; UAS-NPF*[RNAi] (VDRC#108772) is simply referred to as "*TKg>NPF*[RNAi]". Except in Fig. 4b, *Akh-GAL4; UAS-NPFR*[RNAi] (BDSC#25939) is simply referred to as "*Akh>NPFR*[RNAi]".

For adult-specific knockdown of NPF, *TKg-GAL4, tub-GAL80ts (TKg*[ts]) >*LacZ*[RNAi] and *TKg*[ts]>*NPF*[RNAi] flies were raised at 20 °C during the larval, and pupal periods. After eclosion, adult flies were housed at 29 °C for 6 days before experimental analysis.

**Generation of *sut1*[KO] mutant.** The mutant alleles *sut1*[KO] (Fig. 3) was created in a *white* (*w*) background using CRISPR/Cas9 as previously described[36]. The oligo DNA sequences are represented in Supplementary Data 6. The breakpoint detail of *sut1*[KO] is described in Supplementary Fig. 8e.

**Generation of the *UAS-sut1::mVenus* strain and *UAS-sut1* plasmid.** To over-express mVenus-tagged *sut1*, the *sut1* coding sequence region (CDS) was amplified by PCR with adult *w*[1118] whole-body cDNA using the primers sut1cDNA F and sut1cDNA R (Supplementary Data 6), followed by digestion with *EcoRI* (TAKARA) and *XhoI* (TAKARA). *mVenus* CDS was amplified by PCR with a plasmid containing *mVenus* CDS using the primers mVenus cDNA F and mVenus cDNA R (Supplementary Data 6), followed by digestion with *XhoI* (TAKARA) and *NheI* (TAKARA). The digested *sut1* and *mVenus* fragments were ligated with *EcoRI-NheI*-digested pWALIUM10-moe vector[88], leading to *sut1::mVenus*-pWA-LIUM10-moe, which carries two amino acid insertions (leucine and glutamine) between Sut1 and mVenus protein. *sut1::mVenus*-pWALIUM10-moe was then injected into *y*[1] *M{vas-int.Dm}ZH-2A w*\*; *P{y[+t7.7]=CaryP}attP2* embryos[89].

For generation of the *UAS-sut1* plasmid, *sut1* CDS was amplified by PCR with adult *w*[1118] whole-body cDNA using the primers sut1cDNA F and sut1cDNA R2 (Supplementary Data 6). The PCR products were digested with *EcoRI* and *NheI*, and subsequently cloned into the *EcoRI-NheI*-digested pWALIUM10-moe vector.

**Generation of *sut1*[KI-T2A]-GAL4 strain.** We utilised a method previously described[33] to generate a knock-in strain by inserting the T2A-GAL4 cassette into *sut1* locus. Approximately 500 bp sequences flanking the stop codon of *sut1* were PCR amplified from the genomic DNA of the *w*[1118] strain. These homology arms were designed so that T2A-GAL4 was translated as an in-frame fusion with the target protein. The reporter cassette excised from pPG×RF3[33], as well as the left and right homology arms were assembled and cloned into *SmaI*-digested pBlue-scriptII SK(-) in a single enzymatic reaction using the In-Fusion Cloning Kit (TAKARA). gRNA vectors were constructed in pDCC6[90]. We selected a 20 bp gRNA target sequence (Supplementary Data 6) that encompasses the stop codon of the target gene. In addition, silent mutations were introduced into the homology arm of the donor vector to avoid repetitive cleavage after integration. To integrate a reporter cassette into the desired location in the genome, a mixture of a donor vector (150 ng/mL) and a gRNA (150 ng/mL) vector was injected into *yw*[1118] fertilised eggs. After crossing with a balancer strain, transformants in the F1 progeny were selected by eye-specific RFP expression from the 3 × P3-RFP marker gene in adults. The primers used in the generation of *sut1*[KI-T2A]-GAL4 are represented in Supplementary Data 6.

**Antibody preparation.** An antibody against NPF protein was raised in guinea pigs. A KLH-conjugated synthetic peptide (NH2-SNSRPPRKNDVNTMA-DAYKFLQDLDTYYGDRARVRF-CONH2) corresponding to the amidated mature NPF amino acid residues (GenBank accession number NP_536741) were used for immunisation.

**Immunohistochemistry and fluorescence quantification.** Midguts and other fly tissues were dissected in 1× PBS and fixed in 4% paraformaldehyde in PBS for 30–60 min at room temperature (RT). Fixed samples were washed three times in PBS supplemented with 0.1% Triton X-100 (0.1% PBT). The samples were blocked

in blocking solution (PBS with 0.1% Triton X-100 and 2% bovine serum albumin [BSA]) for 1 h at RT and then incubated with a primary antibody in blocking solution at 4 °C overnight. Primary antibodies used in this study were chicken anti-GFP (1:2000, Abcam, #ab13970), rabbit anti-RFP (1:2000, Medical and Biological Laboratories, #PM005), mouse anti-Prospero (1:50; Developmental Studies Hybridoma Bank [DSHB]), guinea pig anti-NPF (1:2000; this study), rabbit anti-Tk (1:2000, a gift from Jan Veenstra)[91], rabbit anti-Bursα (1:1000, a gift from Benjamin H. White)[92], rabbit anti-sNPF (1:1000, a gift from Kweon Yu)[93], rabbit anti-AKH (1:600, a gift from Jae H. Park)[94], rabbit anti-FOXO (1:200, a gift from Marc Tatar)[95], guinea pig anti-DILP2 (1:2000, a gift from Takashi Nishimura)[96], rabbit anti-DILP3 (1:2000, a gift from Jan Veenstra)[91], and rabbit anti-DILP5 (1:1000, a gift from Dick R. Nässel)[97]. After washing, fluorophore (Alexa Fluor 488, 546, 555, or 633)-conjugated secondary antibodies (Thermo Fisher Scientific) were used at a 1:200 dilution, and the samples were incubated for 2 h at RT in blocking solution. After another washing step, all samples were mounted in FluorSave reagent (Merck Millipore).

Midguts samples were dehydrated in a series of ethanol washes ranging from 10% to 90% on ice after fixation in 4% paraformaldehyde. Samples were kept in 90% ethanol for 2 h at −20 °C followed by serial re-hydration and subjected to the staining protocol described above.

Fat bodies were stained with LipidTOX (Thermo Fisher Scientific; 1:1000 in 0.1% PBT) for 2 h at RT after fixation in 4% paraformaldehyde.

Samples were visualised using a Zeiss LSM 700 confocal microscope or Zeiss Axioplan 2. Images were processed using Fiji[98]. Fluorescence intensity in confocal sections was measured via Fiji. We performed the sum-intensity 3D projections to measure total fluorescent intensity across the object of interest (Gut or Brain). For NPF and Bursα quantification, 5–8 cells were examined for each midgut.

**Imaging of glucose sensor**. Ex vivo glucose sensor experiments were performed on dissected midguts. Adult midguts expressing *UAS-FLII12Pglu-700μδ6* were dissected in Schneider's *Drosophila* medium (Thermo Fisher Scientific). The dissected guts were placed on coverslips with 50 μL of Schneider's *Drosophila* medium. Fluorescent images were acquired using a ×40 objective with a Zeiss LSM 700 confocal microscope equipped with the following filter sets: excitation 405 nm, emission 470 nm (CFP channel); excitation 405 nm, emission 530 nm (FRET channel). For calculation of FRET intensity, the FRET ratio (YFP/CFP) was computed using Fiji[98].

To acquire live image data of *Drosophila* S2 cells, S2 cells were seeded in 4 mL of Schneider's *Drosophila* Medium supplemented with 10% heat-inactivated foetal calf serum and 1% penicillin–streptomycin solution (Wako) in a glass bottom dish (IWAKI) 1 day before transfection. S2 cells were transfected using Effectene Transfection Reagent (QIAGEN), as previously described[99] with plasmids for *Actin5C-GAL4* (a gift from Yasushi Hiromi, National Institute of Genetics, Japan) and *UAS-FLII12Pglu-700μδ6*[35] (a gift of Chika Miyamoto and Hubert Amrein), in the presence or absence of the *UAS-sut1* plasmid. Two days after transfection, S2 cells were stored in 3 mL of basal buffer (70 mM NaCl, 5 mM KCl, 20 mM MgCl₂, 10 mM NaHCO₃, 115 mM sucrose, 5 mM HEPES; pH 7.1)[35] for 15 min before experimentation. Next, 1 mL of test solution (basal buffer with 100 mM glucose) was administered through a pipette, bringing the final glucose concentration of cultured medium to 25 mM. Fluorescent images were acquired at ×40 objective using a Zeiss LSM 900 confocal microscope equipped with the following filter sets: excitation 405 nm, emission 470 nm (CFP channel); excitation 405 nm, emission 530 nm (FRET channel). A single fluorescence image frame was acquired every 6 s, and each cell was continuously recorded for 15 min. During this timeframe, the test solution was applied 1 min after recoding began. Images were also analysed by Fiji. An average of 10 frames were obtained before application of the test solution, to define basal FRET levels.

**Quantitative reverse transcription PCR**. To quantify the changes in gene expression, the midguts from 8 to 10 adult female flies, the fly abdomen carcass from ten adult female flies, and the heads from 20 adult female flies were dissected for each sample. For *Akh* mRNA level quantification, 6 whole bodies of adult female flies were sampled. Total RNA was extracted using RNAiso Plus reagent (TaKaRa). cDNA was prepared with ReverTra Ace qPCR RT Master Mix with gDNA Remover (ToYoBo). Quantitative reverse transcription PCR (RT-qPCR) was performed using the Universal SYBR Select Master Mix (Applied Biosystems) with a Thermal Cycler Dice TP800 system (TaKaRa). Serial dilutions of a plasmid containing the open reading frame of each gene were used as standard. The amount of target RNA was normalised to *ribosomal protein 49* (*rp49*) and then relative fold changes were calculated. The primers used to measure transcript levels are represented in Supplementary Data 6.

**Lipid measurement**. Ten flies from each group were homogenised using pellet pestle with 1000 μL PBS containing 0.1% Triton X-100 and heated at 70 °C for 10 min. The supernatant was collected after centrifugation at 17,800 × *g* for 15 min at 4 °C. Ten microliter of supernatant was used for protein quantification using Bradford Reagent (Nacalai tesque). To measure whole-body triglycerides, we processed 10 μL of supernatant using a Serum Triglyceride Determination kit

(Sigma-Aldrich, TR0100). We subtracted the amount of free glycerol from the measurement and then normalised the subtracted values to protein levels.

**CAFÉ assay**. Testing followed a previously published protocol[100]. Four adult virgin female flies were placed in separate tubes (21 mL tube, Sarstedt, 58.489) and two calibrated glass micropipettes (5 μL, VWR) filled with liquid medium (5% sucrose + 5% autolysed yeast extract, Sigma-Aldrich) by capillary action were inserted through the sponge cap. Loss of media due to evaporation was controlled by subtracting readings from identical CAFÉ chambers lacking flies. Liquid media displacement readings were performed manually and divided by four to attain μL/fly/h.

**Haemolymph correction and glucose measurement**. For haemolymph extractions, 30–40 female flies were perforated with a tungsten needle and placed in a 0.5 mL Eppendorf tube perforated with a 27 G needle. The Eppendorf tubes were placed inside 1.5 mL Eppendorf tubes and centrifuged for 5 min at 5000×*g* at 4 °C to collect haemolymph. A 1-μL aliquot of the collected haemolymph was diluted in 99 μL of trehalase buffer (5 mM Tris pH 6.6, 137 mM NaCl, 2.7 mM KCl), followed by heat treatment for 5 min at 70 °C. A 30-μL portion of supernatant was used to measure circulating glucose levels with glucose oxidase assay kit (Sigma-Aldrich, GAGO-20) according to the manufacturer's instructions, as previously described[101]. Trehalose measurement was performed by diluting 30 μL of supernatant with 30 μL of trehalase buffer and 0.09 μL of porcine trehalase (Sigma-Aldrich, T8778-1UN). The solution was then incubated overnight in 37 °C. A 30 μL aliquot of each sample was used to measure circulating trehalose levels with the glucose oxidase assay kit.

**Measurement of circulating DILP2HF level**. The abundance of DILP2 tagged with artificial epitopes (DILP2HF) in haemolymph and whole bodies was measured using a previously described method[52,54]. Briefly, eight-well strips (F8 MaxiSorp Nunc-Immuno modules, Thermo Fisher Scientific, 468667) were incubated at 4 °C overnight with 5 μg/mL anti-FLAG (Sigma-Aldrich, F1804) in 200 mM NaHCO₃ buffer. The eight-well strips were then washed with 0.1% PBT twice and blocked with 4% non-fat skim milk in 0.1% PBT for 2 h at RT. The strips were washed again with 0.1% PBT three times, after which 50 μL of PBS with 0.2% Tween 20 (PBST), containing 25 ng/mL mouse anti-HA antibody conjugated with peroxidase (Roche, 12013819001) and 4% non-fat skim milk, was added to each well. In parallel, ten ad libitum fed 6-day-old flies' abdomens were dissected, submerged in 50 μL of PBST, and gently vortexed for 30 min at RT. After centrifugation of the tubes at 3000 × *g* for 30 s, 50 μL of supernatants were transferred in the prepared eight-well strips (for detection of circulating DILP2HF in haemolymph). After adding 500 μL of assay buffer (PBS with 0.1% Triton X-100 and 4% BSA) to each tube, containing the remaining flies, the flies were grinded using a pestle, and centrifuged at 17,500 × *g* for 1 min at 4 °C. Next, 10 μL of the supernatants were prepared in eight-well strips (for detection of whole-body DILP2HF content). To generate standards for the analysis of circulating DILP2HF levels, a series (0–166 pM) of the synthetic HA::spacer::FLAG peptide standard (NH₂-DYKDDDDKGGGGSYPYDVPDY-CONH₂) was prepared, and 50 μL of standards were transferred into the prepared eight-well strips. Meanwhile, to generate standards for the analysis of whole-body DILP2HF levels, a series (0–829 pM) of the synthetic HA::spacer::FLAG peptide standard was prepared, from which 10 μL of each standard were transferred into the prepared eight-well strips. All mixtures in the eight-well strips were incubated overnight at 4 °C and subsequently washed with 0.1% PBT six times. Next, 100 μL of One-step Ultra TMB ELISA substrate (Thermo Fisher Scientific, 34028) was added to each well and incubated for 15 min at RT; 100 μL of 2 M sulfuric acid was then added to stop the reaction, and absorbance at 450 nm was detected using a plate reader Multikan GO (Thermo Fisher Scientific). The secreted DILP2HF levels were estimated by normalising haemolymph DILP2HF abundance to the whole-body DILP2HF amount. The plates, peptide standard, *UAS-DILP2HF*, anti-FLAG, anti-HA antibody, substrate, and detailed protocol were all generously provided by Seung Kim (Stanford University, USA).

**Starvation analysis**. Adult flies of the desired genotype were collected and aged for 6 days and transferred into 1% agar (in dH₂O) contained in 12 mL vials (SARSTEDT, 58.487). Dead animals were counted in 15, 24, 39, 48, 63, 72, 87, 96 h period. Log rank test or pair-wise log rank test was used to assess statistical significance using R.

**Western blotting analysis**. To quantify the activity of the insulin signalling pathway, the level of AKT phosphorylation (pAkt) was determined by western blotting. For each sample, five adults were homogenised in 150 μL of RIPA buffer with cOmplete protease inhibitor cocktail (Roche) and phosphatase inhibitors (Roche). After centrifugation at 14,000×*g* for 5 min, 75 μL of each supernatant was mixed with 75 μL of 2× Laemmli's loading buffer, and subsequently boiled for 5 min. Next, 7.5 μL of each sample was electrophoresed through a precast 10% polyacrylamide gel (COSMO BIO). Proteins were transferred to a PVDF membrane (Merk Millipore), which was blocked with 5% BSA in PBS containing 0.1% Tween-20 (0.1% PBST) and incubated with rabbit anti-pAkt antibody (Cell

Signalling Technology, 4060S, 1:1000 dilution) or rabbit anti-AKT antibody (Cell Signalling Technology 9272S, 1:1000 dilution) in 5% BSA with 0.1% PBST. Primary antibodies were detected with HRP-conjugated secondary antibodies (GE Healthcare, NA934, NA931), diluted 1:10,000. Signals were then detected using a chemiluminescence method with Lumigen ECL plus (Lumigen) and Ez capture MG (ATTO). After stripping the antibodies by WB Stripping Solution (Nacalai tesque), the membrane was blocked, incubated with mouse anti-β-actin antibody (Santa Cruz Biotechnology, B2008, 1:1000 dilution), and then detected. Full scan images of blot are represented in the Source Data file.

**RNA-seq.** The RNA-seq transcriptional data of adult female carcass obtained from each genotype used for Fig. 2d, and Supplementary Fig. 4 is available from DNA Data Bank of Japan Sequence Read Archive (Accession number DRA010538). For RNA-seq studies, we obtained on average of 30 million reads per biological replicate. We used FASTQC to evaluate the quality of raw single-end reads and trimmed 1 base pair from 3′ end, adaptors and reads of <20q base pairs in length from the raw reads using Trim galore 0.6.4 (Babraham Bioinformatics). Reads were aligned with HISAT2 2.1.0[102] to the BDGP *D. melanogaster* genome (dm6). Next, Samtools 1.9[103] and Stringtie 2.0.6[104] were used to sort, merge, and count reads. The number of trimmed mean of $M$ values (TMM)-normalised fragments per kilobase of combined exon length per one million of total mapped reads (TMM-normalised FPKM value) was calculated with R 3.6.1, Ballgown 2.18.0[104] and edgeR 3.28.0[105,106], and used to estimate gene expression levels. All of the FPKM values and *p*-values corrected with Benjamini–Hochberg false discovery rate (FDR) were presented in Source Data file for RNA-seq.

**Measurement of whole-body and haemolymph metabolites by LC–MS/MS.** Metabolites were measured by using ultra-performance liquid chromatography–tandem mass spectrometry (LCMS-8060, Shimadzu) based on the Primary metabolites package ver.2 (Shimadzu). For whole flies, four samples of five females each were used for each genotype. Whole fly samples were homogenised in 160 μL of 80% methanol containing 10 μM of internal standards (methionine sulfone and 2-morpholinoethanesulfonic acid) and were centrifuged (20,000 × g, 5 min) at 4 °C. Supernatants were de-proteinised with 75 μL acetonitrile, and filtered using 10 kDa Centrifugal Filtration Device (Pall Corporation, OD003C35). After filtration, the solvent was completely evaporated. Haemolymph metabolites were collected from 10 females for each sample. Four samples of each genotype were selected and 115 μL of 100% methanol containing 20 μM of internal standards was added to the haemolymph samples. The protein fraction contained in the haemolymph samples was removed by mixing with chloroform and centrifugation (2300 × g, 5 min) at 4 °C. The supernatant (200 μL) was collected, de-proteinised by adding 100 μL of acetonitrile, and filtered using 10 kDa Centrifugal Filtration Device (Pall Corporation, OD003C35). The solvent was completely evaporated for metabolite analysis. The protein contained in the middle layer was purified by gently mixing with 1 mL of acetone and centrifugation (20,000 × g, 5 min) at 4 °C. This process was repeated two times. After removing acetone, the protein pellet was dried at RT and resolubilised in 50 μL of 0.1 N NaOH by heating for 5 min at 95 °C. The protein amount was quantified by BCA reagent mix (Thermo Fisher Scientific, 23228 and 23224) for normalisation. The evaporated metabolite samples were resolubilised in Ultrapure water (Invitrogen, 10977-023) and injected to LC–MS/MS with PFPP column (Discovery HS F5 (2.1 mm × 150 mm, 3 μm); Sigma-Aldrich) in the column oven at 40 °C. Gradient from solvent A (0.1% formic acid, Water) to solvent B (0.1% formic acid, acetonitrile) were performed during 20 min of analysis. MRM methods for metabolite quantification were optimised using the software (Labsolutions, Shimadzu). The amount of whole-body metabolites was normalised by 2-morpholinoethanesulfonic acid and the body weight, while haemolymph metabolites were normalised by 2-morpholinoethanesulfonic acid and the protein amount.

**Statistics and reproducibility.** All experiments were performed independently at least twice. All immunohistochemical experiments were repeated at least twice with similar results. In each experiment, we analysed three or more specimens. The experiments were not randomised, and the investigators were not blinded. All statistical analyses were carried out using the "R" software environment. The *p*-value is provided in comparison with the control and indicated as * for $p \le 0.05$, ** for $p \le 0.01$, *** for $p \le 0.001$, and "NS" for non-significant ($p > 0.05$). Sample size was determined based on the significance obtained from previous studies with similar experimental setups. Comparable sample sizes were used in each experiment.

**Reporting summary.** Further information on research design is available in the Nature Research Reporting Summary linked to this article.

## Data availability

The raw RNA-seq data generated in this study have been deposited in the DNA Data Bank of Japan Sequence Read Archive database under accession code DRA010538. The raw data generated in this study are provided in the Supplementary Information/Source Data file. Source data are provided with this paper.

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

## Acknowledgements

We thank Hubert Amrein, Yasushi Hiromi, Seung Kim, Ronald P. Kühnlein, Masayuki Miura, Chika Miyamoto, Dick R. Nässel, Takashi Nishimura, Jae H. Park, Norbert Perrimon, Yi Rao, Ping Shen, Marc Tatar, Jan Veenstra, Benjamin H. White, Daisuke Yamamoto, Kweon Yu, the Bloomington Stock Center, the Kyoto Stock Center (DGRC), the National Institute of Genetics, the Vienna Drosophila Resource Center, and the Developmental Studies Hybridoma Bank for providing stocks and reagents; and Takefumi Kondo, Yukari Sando, and Tadashi Uemura for their technical support of the next-generation sequencing. Y.Y. and H.K. were recipients of the fellowship from the Japan Society for the Promotion of Science. F.O. was a TARA Project Investigator of University of Tsukuba. This work was supported by grants from AMED-CREST, AMED (21gm1110001h0005) to R.N., AMED-PRIME, AMED (21gm6310011h9902) to F.O., and KAKENHI (26250001 and 17H01378 to H.T., 18J20572 to Y.Y., and 19H03367 to – F.O.). This was also supported by the Joint Usage/Research Center for Developmental Medicine, IMEG, Kumamoto University. We would like to thank Editage (www.editage.com) for English language editing.

## Author contributions

Y.Y. and R.N. designed and conceived the study. Y.Y. performed most of the experiments and analysed data. T.K. assisted with RNA-seq analysis. R.H. conducted some immunohistochemistry. R.M. established *sut1* genetic mutant. H.K. and F.O. performed metabolomic analysis. S.K. and H.T. established *NPF* mutant, *NPFR* mutant, and *NPFR*^{T2A}-*GAL4* strain. A.N. assisted with the generation of *sut1*^{T2A}-*GAL4* strain. Y.Y. and R.N. wrote the manuscript.

## Competing interests

The authors declare no competing interests.
