## [Peer Review File · Nature Communications]

REVIEWER COMMENTS

Reviewer #1 (Remarks to the Author):

This interesting paper from Yoshinari et al presents a survey of the physiological effects and pathway function of Neuropeptide F (NPF) as regulator of sugar metabolism and fat catabolism in *Drosophila*. The authors present extensive data indicating that NF produced in enteroendocrine cells (EECs) in the gut in response to a sugar diet controls the secretion of AKH and DILPs from the fly's corpora cardiaca (CC) and insulin producing cells (IPCs), respectively. AKH and DILP then regulate metabolism in the Fat Body via FOXO, suppressing lipolysis and promoting sugar and fat storage. This mode of regulation makes intuitive sense, and is especially interesting because the way NF is utilized closely parallels incretin function in mammals (see Fig 9). The experimental treatment is quite comprehensive, using ligand and receptor knockouts in the various organs combined with relevant genomics and metabolic assays. Although many of the effects that are documented are rather small in magnitude, the data are nevertheless generally convincing. The genetics are first rate, utilizing elegant rescue experiments in many cases. Moreover, the paper is well written and quite clear. In the very active field of organ-organ communication and metabolic control in *Drosophila*, this paper looks like a significant advance. I have a few minor comments about presentation, experimental design, and data quality (below), but overall I'm quite enthusiastic about this work.

Specific comments:

1. (Line 88): Please indicate where else TKg-Gal4 is expressed, if anywhere.
2. There are a few grammatical errors that should be corrected (Lines 117, 49, 230, 291, 300).
3. Line 128: please indicate if whole animal RNAseq was used.
4. In several instances the paper refers to gene "expression" when discussing mRNA levels measured by RT-qPCR. Please indicate when mRNA is assayed, and when protein is assayed, in both the text and the figures.
5. Line 137: note that this analysis was in Fat Body.
6. Lines 146-147: please comment on the size and morphology of the Fat Body. Were there notable alterations? If so, show these in a figure please.
7. Figure 3, Lines 156-160: The authors find increased NPF protein in EECs after starvation, and less NPF after a sugar feeding. They conclude from this that NPF secretion is promoted by sugar feeding. This inference is critical to the authors' model of NPF function, but their conclusion is only one of several alternatives that could be taken from the observations. This is one of the weakest links in the story, and it would be good to see more data to support the authors' conclusion about NPF secretion. Measurements of hemolymph NPF would be ideal if these are possible. The same issue applies to DILP expression and secretion (line 284) and here again, direct data on secretion or hemolymph levels should be provided if possible. Technically, I imagine this is quite challenging.
8. The RNAseq data shown in Figure 2a and 2b are not conclusive, and can be shown in the supplement.
9. In Fig 2, it would be good to show Trehalose levels, since these are the main circulating sugar in fly hemolymph.
10. A bit more data on the expression patterns of the NPFR-Gal4 lines needs to be included. Where exactly is this receptor expressed? Please provide a complete description, and more evidence in the supplement.
11. Figure 5a is another major weak link in the story. Loss of NPFR in the CC reduces Akh mRNA there, but only a little bit and the difference is not very significant. More data should be included to support the function of NPF and NPFR as regulators of Akh, if possible. The genetic analysis on this is great (Fig 5b-k), but ideally the authors would also have measures of Akh protein, or more direct measures of its activity.
12. The authors should quantify the tGPH activity in Fig 7e, if possible. A ratiometric assay of membrane/cytoplasmic tGPH would be a good addition here.
13. Fig 9 has a typo in it ("corpola cardiaca"). I also suggest altering this figure to include "lipid

storage" as an effect of DILPs, and "Lipolysis" as an effect of Akh. The authors might also consider combining Fig 9 with Fig 8h. These summary figures are very helpful for guiding the reader through the data, and they should be included in the main body of the paper.

Reviewer #2 (Remarks to the Author):

This is an interesting study linking the function of enteroendocrine cells (EECs) with general energy homeostasis in the *Drosophila* model. The authors propose that NPF (the invertebrate NPY homolog) produced by EECs serves as an "incretin-like" hormone, controlling both the production of AKH (a glucagon-like hormone) and insulin-like peptides, therefore modulating fat and sugar homeostasis in adult flies.

The study is rather complete and presents numerous experimental approaches to demonstrate these functional links. However, several major links are missing, among which the demonstration that NPF produced by the EECs is secreted and that NPF derived from EECs, and not from other sources, modulates NPFR signaling in the corpora cardiaca and the Insulin-producing cells (IPCs).

Experimental evidence for these two points are mandatory to support the conclusions of the paper.

Major remarks:

1- Fig2a: if the metabolic changes in TKg>NPFi are not significant, the conclusion should be differently written (the "trend" has no significance, then), and the data should not be part of a main figure. Line 130, it is improper to write "...were also...", since it suggests that the previous data demonstrates a change in regulation, which is not the case. What is the statistical significance of the data presented in Fig2b?

2- Fig3: accumulation of NPF in the EECs upon starvation suggests a defect in secretion, but is certainly not a proof. This is an important caveat of the manuscript and the authors should measure circulating levels of NPF in starved vs control food conditions. This should also be done in TKg>NPF-RNAi conditions to ensure that NPF production by the EECs majorly contributes to its circulating levels in the hemolymph. This is particularly important since NPF is also produced by specific neurons in the brain, which could equally contribute to its circulating levels.

In that respect, the authors should also assess the accumulation of NPF in brain neurons upon starvation and measure circulating levels after knock-down of NPF in these neurons in fed conditions.

3- Experiments using Sut1 knock-down suggest that this transporter is required for normal sugar/fat homeostasis. However, there is no evidence that Sut1 transports glucose into cells, as hypothesized by the authors. This should be assessed experimentally.

TKg>Sut1-RNAi only partly mimics the effect of starvation on NPF accumulation. Possible explanation and their experimental testing should be provided. In fact, according to fig. 3h, the difference between sut1+/- and sut1-/- is not significant, therefore questioning the result obtained with the unique sut1-RNAi line used.

4- Fig4a (and Suppl. 4c): why is the co-staining between GFP and AKH only partial if GFP expression is targeted to the CCs?

Suppl.Fig.4f: the absence of direct connection between NPF+ neurons and the CCs does not prove that these neurons cannot communicate with the CCs, since they could contribute to circulating PDF. This should be modified in the text and tested experimentally (see point 2).

5- p7 line 261 "...suggesting that...". This conclusion seems wrong. Bmm expression is upregulated by Akh>NPFRi, but not dHSL expression. This suggests that dHSL function is not controlled (at least transcriptionally) by NPFR signaling in the CC. Silencing dHSL expression in fat cells could rescue NPF loss-of-function starvation sensitivity by a parallel mechanism, without implying that dHSL function relies on NPFR.

6- p12 line 289: The functional link between NPFR in the IPCs, and NPF produced from the EECs is not established. Therefore, such conclusion should not be made. Again, NPF could be secreted from the central brain and activate NPFR in the IPCs.

7- Fig7d is not convincing. It is difficult to imagine that such a mild reduction in IPC neuronal activity could account for a notable difference in Dilp secretion.

8- Fig7e lacks quantification and confirmation with another marker for insulin signaling. As such this piece of data is not significant.

9- Throughout their manuscript, the authors present a set of related phenotypes due to EEC-specific NPF knockdown consisting in: resistance to starvation, reduced lipid accumulation, increased feeding and reduced glycemia. Experimental evidence suggest that these effects are due to NPF sending metabolic signals to the CC and the IPCs, AKH and Dilps hormones acting then as relays. However, the phenotypes observed after silencing NPFR in the CC and IPCs are not consistent with this idea and it is unclear what contribution NPFR in the IPCs and the CCs brings to the NPF knockdown phenotype. Strikingly, Dilp2>NPFR-RNAi animals show reduced TAG amount, glycemic levels and increased food intake, but normal resistance to starvation (Figure 8a). This contrasts with the strong decrease in resistance to starvation observed in AKH>NPFR-RNAi animals (Figure 5b) and indicates that this phenotype is controlled by AKT only. It also indicates that resistance to starvation is independent of decreased lipid accumulation and glycemia, and increased feeding. The authors should provide an explanation for these discrepancies and mention it in the text.

Minor remark

1- The fly food recipe used should be detailed for the reader, since many metabolic phenotypes have been shown to specifically rely on the equilibrium between food components.

Reviewer #3 (Remarks to the Author):

In this manuscript, Yoshinari et al. report that midgut-derived NPF may act as a sugar-responsive incretin-like hormone in *Drosophila*. Similar to the regulation of glucagon and insulin by incretin in mammals, the authors show that NPF is released in response to sugar and regulates the counterparts of glucagon and insulin in the fruit fly, Akh and Dilps, by suppressing the secretion of the former and promoting the secretion of the latter. NPF could be the first functional homolog of incretin identified in the invertebrate. The model proposed by the authors is very appealing and shows the deep homology in the principles of metabolic regulation between the vertebrate and invertebrate.

Although, overall, the work is interesting and significant, there are some concerns need to be addressed to make the conclusions more convincing.

Major concerns:

1, The authors showed solid data that all the phenotypes induced by knocking-out or knocking-down NPF (in Figure 1) were caused by the deficiency of NPF. However, based on the current evidence, it is not very convincing to claim that these phenotypes were caused specifically/exclusively by NPF secreted from the midgut EECs. Considering the humoral nature of NPF, it is possible that the decrease of NPF peptides in the circulating system is sufficient to lead to these phenotypes, regardless the origin of the decrease. For example, brain NPF secretion may also be tightly regulated by diet, in that there are also sugar/nutrient sensors in the brain. The following efforts may help address this concern.

First, test if knocking down NPF in other tissues, especially in the central nervous system, causes the same phenotypes.

Second, test if restoring NPF in other tissues, especially in the central nervous system, by either NPF overexpression in NPF neurons or optogenetic/themogenetic activation of NPF neurons, could rescue the phenotypes seen in NPF mutant. The neural-specific manipulation could be achieved by more restrictive GAL4 lines or *otd-flp*.

Third, simply discuss the alternative possibilities in the discussion.

2, The evidence for the connection between NPFR and IPCs neural activity is not very convincing.

First, the NPFR couples to a Gi signaling pathway to inhibit NPFR-expressing neurons.

Second, the quantifications in Figure 7d lack an internal control, and Figure 7e does not have quantified group data. For example, in Figure 7e, the authors could use DAPI signal as an internal control to normalize the tGPH signals. Actually, the DAPI signal in Figure 7e lower panel (NPFR-RNAi group) appears dimmer than the control groups, suggesting the change in the tGPH signals may just be an artifact due to lower background in that particular staining. In Figure 7d, the authors could also use treatments, such as starvation and re-feeding, to achieve more reliable CaLexA signals, or they could use a similar assay, the TRIC assay, which has RFP expression as an internal expression control.

Minor concerns:

1, Is there a bi-directional regulation of Akh and Dilps by NPF/NPFR? Would opposite phenotypes (in starvation resistance or TAG level) be observed if NPF/NPFR is overexpressed in a wildtype genetic background?

2, The authors may include the phenotypes of *Akh>Akh-RNAi* or *Akh-A/Akh-KO* in some of the experiments in Figure 5. It would further confirm that it is Akh in the Akh cells that contributes to the phenotypes. It would also provide clues about whether the phenotypes are bi-directionally regulated. Along the same line, the authors may test if the phenotypes of *Dilp2>Dilp2/3/5-RNAi* recapitulate the effect of knocking-down NPFR in IPCs to further support the model in Figure 8h.

3, The authors indicated that the behavioral and metabolic phenotypes induced by knocking-down NPFR in the IPCs is caused by attenuated insulin signaling. I think it is a bit of a stretch to attribute both phenotypes to insulin signaling. Feeding and metabolism may be regulated by different mechanisms in the IPCs. For example, most IPCs produce Dilp2/3/5 as well as DSK. It has been shown in previous studies that *Dilp2>DSK-RNAi* flies also show increased food intake and starvation resistance. It is not rigorous to say Dilps, regulated by NPFR, play the role in both functions without excluding the potential contribution by DSK.

4, The authors consistently observed that the expression level and the protein level of a peptide (NPF and Dilps) changed in opposite directions. It would be better if more discussion could be made about potential mechanisms for this phenomenon. Along this line, is there any reason that the authors only showed the expression level of Akh but not the protein level?

Reviewer #4 (Remarks to the Author):

This manuscript describes incretin-like roles for gut-derived Neuropeptide F (NPF, homologous to mammalian NPY) in adult *Drosophila*. The same lab had previously shown that gut NPF promotes

germline stem cell proliferation in mated females (PMID: 30248087) and they now explore its metabolic roles (in mated females too).

The authors show that:

1. Constitutive loss of gut enteroendocrine (EE) cell-derived NPF leads to starvation-sensitive flies. These flies have reduced triacylglyceride and a transcriptional/metabolic profile suggestive of a starvation-like state, despite eating more.
2. The levels of NPF within EE cells are sensitive to dietary sugars and dependent on the sugar transporter1 (*sut1*) gene
3. In the fed state, NPF signals through its receptor to the secretory cells that make glucagon-like Akh hormone to prevent Akh release; a compelling experiment in support of this idea is that loss of Akh can rescue the lipodystrophy resulting from NPF receptor mutation or its specific downregulation in Akh-producing cells
4. In the fed state, NPF promotes insulin release, and the NPF effects on Akh and insulin collectively regulate peripheral FOXO levels, sugar and lipid metabolism.

These conclusions are generally supported by comprehensive experiments, and the data is of very good quality. I have three general comments, and a few specific ones:

1. The roles of NPF are practically identical to those recently described for another hormone co-expressed in the same EE cells: Bursicon alpha (*Bur*, PMID: 30344016). This is fine, but the claims about NPF being the first incretin should be a bit toned down. More importantly, the two manuscripts together raise a few questions about how the same EE cells use different mechanisms to detect the same nutrient (sugar), resulting in potentially differential release of two different EE peptides that end up doing the same thing. This does not make sense to me and needs to be addressed somehow. Specifically, does *Sut1* regulate *Bur* expression? Is NPF not sensitive to *Glut1*, previously implicated in the sugar-stimulated secretion of *Bur*s from EE cells? Is *Sut1* really acting in EE cells? The only data in support of this idea one single EE-specific *sut1* RNAi downregulation, and *Sut1* expression in EE has not been shown. This is important because, even though the *TKg-Gal4* driver used for this downregulation is meant to be gut-specific, it is actually expressed in a small subset of neurons, so the reported phenotypes could be due to sensory/central neuronal sugar sensing/transport.
2. The authors make many statements about NPF secretion without directly demonstrating that NPF is, in fact, secreted as an incretin. I realise that it is not trivial to measure circulating hormones in flies, but this has been achieved for *Bur*s made by the same cells (PMID: 30344016) using Western blots. Do the levels of circulating NPF decrease when NPF is downregulated from EEs and/or in response to sugar restriction?
3. As far as I can tell, all genetic manipulations were performed constitutively throughout development (*TKg-Gal4* is expressed in third instar larvae possibly earlier). This makes some phenotypes difficult to interpret, such as those affecting fat body lipid and starvation sensitivity; an inability to mobilise fat stores might be expected to result in increased rather than decreased fat stores. To resolve this, the authors should test whether acute depletion of NPF in adult EE cells (by means of a *tub-Gal80ts* transgene for example) leads to reduced lipid stores, starvation sensitivity and hyperphagia.

Specific comments

1. Lines 66-67. Not sure it is appropriate to describe *Activin β* as an enteroendocrine hormone?
2. Lines 66-68. Please provide relevant references.
3. Line 162: "we hypothesised that starvation impairs NPF secretion from EECs". Consider rephrasing:

"impairs" somehow implies a failure when it is in fact the adaptive, homeostatic response.

4. Line 214: "re-introduction" may be more accurate than "overexpression" given that the manipulation is conducted on a mutant that does not express NPFR.

5. There are a few typos in some figure headings and the summary cartoon.

6. The visceral muscle expression of the NPFR knockin line is surprising in light of its previously reported expression (PMID: 11897397). Although that study used larval rather than adult guts, expression appears to be in epithelial precursors/enteroendocrine cells. The two knockin lines used to assess NPFR expression also seem to differ in the CNS: one looks almost pan-neuronal which seems unlikely. Expression analysis of the endogenous NPFR transcript or protein would increase my confidence in the reported expression.

7. The authors may want to mention Limostatin (a fly decterin, PMID: 25651184) in their Discussion, because together, the two studies suggest that an incretin may be regulating secretion of a decterin.

8. The raw data needs to be provided for metabolomics/transcriptomics datasets.

9. It seems interesting that NPF levels (both protein and transcript) are even higher in peptone-refed than normally fed animals. Might this suggest some integration of sugar/protein amounts by EE cells? The authors might want to discuss this.

10. I believe the primary reference for the Ilp3 antibody is PMID: 18972134.

11. My understanding is that most experiments were done in mated females, but the CAFÉ assay used virgin females. Please clarify/justify.

Response to Referees

Re: MS# NCOMMS-20-29096-A

‘The sugar-responsive enteroendocrine neuropeptide F regulates lipid metabolism through glucagon-like and insulin-like hormones in *Drosophila melanogaster*’

(The original title ‘The enteroendocrine neuropeptide F acts as an incretin-like hormone in *Drosophila*’)

Dear Editor and Reviewers,

We would like to thank you and the reviewers for your comments, which have strengthened our manuscript. The reviewers’ concerns are reproduced below, and our responses are presented in **bold**. All of these changes can be tracked in the revised text by **yellow markers**. We hope that this revised manuscript is now suitable for publication in *Nature Communications*.

Sincerely,

Ryusuke Niwa

Response to Reviewer #1

This interesting paper from Yoshinari et al presents a survey of the physiological effects and pathway function of Neuropeptide F (NPF) as regulator of sugar metabolism and fat catabolism in *Drosophila*. The authors present extensive data indicating that NF produced in enteroendocrine cells (EECs) in the gut in response to a sugar diet controls the secretion of AKH and DILPs from the fly's corpora cardiaca (CC) and insulin producing cells (IPCs), respectively. AKH and DILP then regulate metabolism in the Fat Body via FOXO, suppressing lipolysis and promoting sugar and fat storage. This mode of regulation makes intuitive sense, and is especially interesting because the way NF is utilized closely parallels incretin function in mammals (see Fig 9). The experimental treatment is quite comprehensive, using ligand and receptor knockouts in the various organs combined with relevant genomics and metabolic assays. Although many of the effects that are documented are rather small in magnitude, the data are

nevertheless generally convincing. The genetics are first rate, utilizing elegant rescue experiments in many cases. Moreover, the paper is well written and quite clear. In the very active field of organ-organ communication and metabolic control in *Drosophila*, this paper looks like a significant advance. I have a few minor comments about presentation, experimental design, and data quality (below), but overall I'm quite enthusiastic about this work.

Specific comments:

1. (Line 88): Please indicate where else TKg-Gal4 is expressed, if anywhere.

Response: This data was published in our previous work (Ameku et al. *PLOS Biol.* 2018). In the revised manuscript, we have cited Ameku et al. (Ref #17) and mentioned the expression pattern as follows: “This *GAL4* driver is active in most NPF+ EECs and small subsets of neurons but not in NPF+ neurons.” (P5, L99–100).

2. There are a few grammatical errors that should be corrected (Lines 117, 49, 230, 291, 300).

Response: We have revised the text as per your comments (P4, L60; P13, L304; P16, L372). Additionally, the text appearing on L117 and L300 in the original manuscript have been deleted in the revised version.

3. Line 128: please indicate if whole animal RNAseq was used.

Response: As the reviewer suggested, we have indicated “the abdomens of adult females” in the revised manuscript (P8, L165). We have also indicated ‘abdomen’ in other sites of the revised manuscript when necessary.

4. In several instances the paper refers to gene "expression" when discussing mRNA levels measured by RT-qPCR. Please indicate when mRNA is assayed, and when protein is assayed, in both the text and the figures.

Response: As suggested by the reviewer, we have clearly indicated whether it was the mRNA or protein being referenced throughout the manuscript and figures, such as ‘mRNA expression.’

5. Line 137: note that this analysis was in Fat Body.

Response: This comment is related to comment #3. Our samples were not derived from the fat body but rather the abdomen of adult females. We have clarified this point in the revised manuscript (P8, L165 and others).

6. Lines 146-147: please comment on the size and morphology of the Fat Body. Were there notable alterations? If so, show these in a figure please.

Response: There are no visible alterations in the size or morphology of the fat body, while lipid droplet intensity appeared reduced. We have added a lower magnification photo showing the morphology of the fat body (Fig. 1i) and have described this point on P7, L135–136 in the revised manuscript.

7. Figure 3, Lines 156-160: The authors find increased NPF protein in EECs after starvation, and less NPF after a sugar feeding. They conclude from this that NPF secretion is promoted by sugar feeding. This inference is critical to the authors' model of NPF function, but their conclusion is only one of several alternatives that could be taken from the observations. This is one of the weakest links in the story, and it would be good to see more data to support the authors' conclusion about NPF secretion. Measurements of hemolymph NPF would be ideal if these are possible. The same issue applies to DILP expression and secretion (line 284) and here again, direct data on secretion or hemolymph levels should be provided if possible. Technically, I imagine this is quite challenging.

Response: We appreciate your critical question regarding measurements of hemolymph NPF. Over the last 8 months and half, since receiving the reviewers' feedback, we have tried several experiments (ELISA, Dot blotting, western blotting, and LS-MS/MS analysis) to address this point. However, to date, we have been unsuccessful in detecting the circulating NPF levels in *Drosophila* hemolymph samples. We assume that this is due to the low abundance of endogenous NPF in the hemolymph, or its high rate of degradation in the hemolymph. As the reviewer noted, this is a quite challenging issue, and we cannot further try to promptly develop an appropriate detection method in the COVID-19 pandemic in Japan. I hope

that the reviewer will understand our situation. We have, however, addressed this issue in the Discussion section of the revised manuscript (P20, L467–470). In addition, we have mentioned that we ‘hypothesize’ that starvation suppresses NPF secretion from EECs (P9, L205 – P10, L206).

Regarding DILP level in the hemolymph, we quantified DILP2 protein abundance in the hemolymph upon *NPFR* RNAi in the IPCs using a method with the DILP2HF strain, as previously reported (Park et al. *PLOS Genetics* 2014; Heshan et al. *Nature Communications* 2018). The methodology is described in P32, L768 – P33, L798. We found that knockdown of *NPFR* in the IPCs significantly decreased hemolymph DILP2 protein level, which is consistent with the reduction in the IPC activity (Fig. 7d). These data are described on P17, L381–385.

8. The RNAseq data shown in Figure 2a and 2b are not conclusive, and can be shown in the supplement.

Response: As the reviewer suggested, we have moved the data to the supplemental figure (Extended Data Fig. 4a, b). We have also moved a portion of the metabolomic data from the supplemental material to the main figure (Fig. 2a, b)

9. In Fig 2, it would be good to show Trehalose levels, since these are the main circulating sugar in fly hemolymph.

Response: Thank you for this important suggestion. We measured circulating trehalose levels of *TKg^{ts}>NPFR^{RNAi}* animals, and confirmed that knockdown of gut *NPF* resulted in the reduction of not only glucose but also trehalose in the hemolymph. We have described this data in Extended Data Fig. 2b and P7, L136–137.

10. A bit more data on the expression patterns of the *NPFR*-Gal4 lines needs to be included. Where exactly is this receptor expressed? Please provide a complete description, and more evidence in the supplement.

Response: We have added a comprehensive atlas for the expression pattern of *NPFR^{T2A-KI}-GAL4* in Extended Data Fig. 11 and on P12, L276 –

P13, L282. We determined that *NPFR^{T2A-KI}-GAL4* is expressed in the brain, CC, sNPF+ enteric neurons, Malpighian tubules, ovary, and gut.

11. Figure 5a is another major weak link in the story. Loss of NPFR in the CC reduces Akh mRNA there, but only a little bit and the difference is not very significant. More data should be included to support the function of NPF and NPFR as regulators of Akh, if possible. The genetic analysis on this is great (Fig 5b-k), but ideally the authors would also have measures of Akh protein, or more direct measures of its activity.

Response: Thank you very much for raising this critical point. In response to this comment, we have confirmed AKH protein abundance in the CC by immunohistochemistry. Compared to the control, midgut-specific *NPF* knockdown (*TKg>NPF^{RNAi}*) and CC-specific *NPFR* knockdown (*Akh>NPFR^{RNAi}*) decreased AKH protein in the CC (Fig. 5b, c). In conjunction with the observed upregulation of *Akh* expression, these data suggest that loss of NPF/NPFR signalling promotes AKH secretion from the CC, leading to activation of AKH/AKHR signalling. This point is described on P13, L305 – P14, L307.

12. The authors should quantify the tGPH activity in Fig 7e, if possible. A ratiometric assay of membrane/cytoplasmic tGPH would be a good addition here.

Response: As the reviewer suggested, we performed a ratiometric assay to detect membrane/cytoplasmic tGPH signals and have added the corresponding data in Fig. 7f. The quantitative data further supports our conclusion that tGPH signalling at the plasma membrane of the fat body is significantly reduced in *Dilp2>NPFR^{RNAi}* animals.

13. Fig 9 has a typo in it ("corpola cardiaca"). I also suggest altering this figure to include "lipid storage" as an effect of DILPs, and "Lipolysis" as an effect of Akh. The authors might also consider combining Fig 9 with Fig 8h. These summary figures are very helpful for guiding the reader through the data, and they should be included in the main body of the paper.

Response: We have fixed the typographical errors and have revised the phrased related to Fig. 9, in response to the reviewer's comments. In

addition, as the reviewer recommended, we have combined the original Fig. 8h with Fig. 9 in the main body of the revised manuscript.

Response to Reviewer #2

This is an interesting study linking the function of enteroendocrine cells (EECs) with general energy homeostasis in the *Drosophila* model. The authors propose that NPF (the invertebrate NPY homolog) produced by EECs serves as an “incretin-like” hormone, controlling both the production of AKH (a glucagon-like hormone) and insulin-like peptides, therefore modulating fat and sugar homeostasis in adult flies.

The study is rather complete and presents numerous experimental approaches to demonstrate these functional links. However, several major links are missing, among which the demonstration that NPF produced by the EECs is secreted and that NPF derived from EECs, and not from other sources, modulates NPFR signaling in the corpora cardiaca and the Insulin-producing cells (IPCs). Experimental evidence for these two points are mandatory to support the conclusions of the paper.

Major remarks:

1- Fig2a: if the metabolic changes in TKg>NPFi are not significant, the conclusion should be differently written (the “trend” has no significance, then), and the data should not be part of a main figure. Line 130, it is improper to write “...were also...”, since it suggests that the previous data demonstrates a change in regulation, which is not the case. What is the statistical significance of the data presented in Fig2b?

Response: We would like to thank you for your constructive comments. As you have pointed out, the expression changes of certain genes shown in the original Fig. 2a (as well as the original Fig. 2b) were not significant. This figure includes most of the curated genes related to carbohydrate metabolism and TCA cycle regardless of statistical significance. We agree with the reviewer’s opinion, and therefore, have moved the data shown in the original Fig.2a and 2b to Extended Data Fig. 4a, b. We have also carefully revised the text describing these data on P8, L164–173. In the revised text, we have specified that statistic significant was set at $p < 0.05$, which is described in Extended Data Tables 1 and 2.

2- Fig3: accumulation of NPF in the EECs upon starvation suggests a defect in secretion, but is certainly not a proof. This is an important caveat of the manuscript and the authors should measure circulating levels of NPF in starved vs control food conditions. This should also be done in TKg>NPF-RNAi conditions to ensure that NPF production by the EECs majorly contributes to its circulating levels in the hemolymph. This is particularly important since NPF is also produced by specific neurons in the brain, which could equally contribute to its circulating levels.

In that respect, the authors should also assess the accumulation of NPF in brain neurons upon starvation and measure circulating levels after knock-down of NPF in these neurons in fed conditions.

Response: We would like to thank you for raising this important point. Over the last 8 months and half, since receiving the reviewers' feedback, we have tried several experiments (ELISA, Dot blotting, western blotting, and LS-MS/MS analysis) to address this point. However, to date, we have been unsuccessful in detecting the circulating NPF levels in *Drosophila* hemolymph samples. We assume that this is due to the low abundance of endogenous NPF in the hemolymph, or its high rate of degradation in the hemolymph. In the COVID-19 pandemic in Japan, we cannot further try to promptly develop an appropriate detection method. I hope that the reviewer will understand our situation. We have, however, addressed this issue in the Discussion section of the revised manuscript (P20, L467–470). In addition, we have mentioned that we 'hypothesize' that starvation suppresses NPF secretion from EECs (P9, L205 – P10, L206).

Additionally, rather than directly measuring the hemolymph NPF level, we assessed the contribution of brain NPF with a *fbp-GAL4*, which is active in the brain *NPF*⁺ neurons, but not in gut EECs. In contrast with the knockdown of *NPF* in the midgut, knockdown of *NPF* in the brain did not have significant effects on starvation resistance, TAG level, or feeding (Extended Data Fig. 3a-g). Moreover, knockdown of brain *NPF* did not alter Dilp2,3-5 levels or AKH levels (Extended Data Fig. 15a-d). We also confirmed that starvation does not induce accumulation of NPF in the brain (Extended Data Fig. 15g). These data suggest that brain NPF plays a distinct role with gut NPF in regulating metabolism. These data are described on P7, L149 – P8, L161 and P19, L437–446.

3- Experiments using Sut1 knock-down suggest that this transporter is required for normal sugar/fat homeostasis. However, there is no evidence that Sut1 transports glucose into cells, as hypothesized by the authors. This should be assessed experimentally.

TKg>Sut1-RNAi only partly mimics the effect of starvation on NPF accumulation. Possible explanation and their experimental testing should be provided. In fact, according to fig. 3h, the difference between *sut1*^{+/-} and *sut1*^{-/-} is not significant, therefore questioning the result obtained with the unique *sut1*-RNAi line used.

Response: Thank you very much for providing us with this important critique. To address your first question, we conducted the following three experiments.

(1) We utilized *Drosophila* S2 culture cells expressing *Glu*⁷⁰⁰, a FRET-based glucose sensor (Takanaga et al. *Biochim. Biophysic. Acta*, 2008; Volkenhoff et al. *J. Insect Physiol.*, 2018) and found that *sut1* overexpression in S2 cells enhanced the *Glu*⁷⁰⁰ FRET signal as compared to a negative control (no *sut1* overexpression) after glucose was added to the medium (Extended Data Fig. 7a). This result suggests that Sut1 mediates the uptake of extracellular glucose into S2 cells. The result has been described on P11, L233–239 in the revised manuscript. The experimental procedure is described on P30, L707–724.

(2) We found that the *Glu*⁷⁰⁰ FRET signal was reduced in *sut1*-knockdown EECs as compared to control EECs. In addition, 24-hour starvation significantly decreased FRET signals in control EECs (Extended Data Fig. 7b). This data suggests that *sut1* regulates intracellular glucose levels in EECs, which depends on dietary nutrients. These results have been described on P11, L239–244 in the revised manuscript. The experimental procedure is described on P29, L700 – P30, L706.

(3) We established a new strain to overexpress mVenus-tagged Sut1 protein (Sut1::mVenus) and confirmed that Sut1::mVenus was localized on cellular plasma membranes (Fig. 3e), consistent with the postulate that Sut1 is involved in the transport of its *bona fide*

substrate(s). These results have been described on P10, L230 – P11, L232 in the revised manuscript.

It should, however, be noted that we have not directly assessed the transport activity or affinity of Sut1 for glucose. While future studies should investigate this point, we believe that this is beyond the scope of the current manuscript.

Regarding the second point that the reviewer raised, we agree that the *sut1*^{-/-} mutant did not show increased NPF protein in the gut, whereas *NPF* transcript level was significantly decreased (Fig. 3g). This situation was also observed in *sut1* knockdown using another *UAS-sut1^{RNAi}(TRiP)* line. However, *TKg>Sut1^{RNAi}(VDRCKK)* exhibited increased NPF protein in the EECs. Due to these inconsistencies in the data, we have moved the data showing NPF protein levels to the Extended Data from the main body (Extended Data Fig. 6d-e, Extended Data Fig. 8b, f). In contrast, we would like to emphasize that the reduction of *NPF* mRNA levels with upregulation (or trended upregulation) of NPF protein level are consistent with the results obtained from the three loss of *sut1* experiments (Fig. 3f, g, Extended Data Fig. 8a). Therefore, we surmise that loss of *sut1* in the EECs resulted in suppression of NPF signalling in peripheral tissues. This hypothesis is also supported by metabolic data for *TKg>sut1^{RNAi}* animals (Fig. 3h-j, Extended Data Fig. 8c, d).

4- Fig4a (and Suppl. 4c): why is the co-staining between GFP and AKH only partial if GFP expression is targeted to the CCs?

Suppl.Fig.4f: the absence of direct connection between NPF+ neurons and the CCs does not prove that these neurons cannot communicate with the CCs, since they could contribute to circulating PDF. This should be modified in the text and tested experimentally (see point 2).

Response: Thank you for this important criticism. The cells that the reviewer mentioned are not CC cells but rather neurons expressing the neuropeptide short NPF (sNPF). To clarify this point, we conducted co-immunostaining with anti-AKH antibody and anti-sNPF antibody and confirmed that the *NPFR-T2A-KI-GAL4⁺* cells near the CC were labelled with the anti-sNPF antibody, but not with the anti-AKH antibody (Extended

Data Fig. 11c). This result suggests that the cells are sNPF-positive neurons. Since the cell morphology of sNPF neurons is similar to enteric neurons, we concluded that *NPFR-T2A-KI-GAL4* is also expressed in the enteric neurons. We have addressed this point in the main text (P11, L278–279) and in the figure legend of Extended Data Fig. 11c.

As per the second point raised by the reviewer regarding the reliability of the trans-Tango experiment, we have modified the text (P19, L443–444) and removed the statement “NPF neurons do not have a direct connection with the CC and that circulating NPF in hemolymph may be received in the CC cells”. To assess the involvement of brain NPF for AKH regulation, we conducted brain-specific knockdown of *NPF* and found that the knockdown had no significant effects on *AKH* mRNA and protein levels (Extended Data Fig. 15c, d, and P19, L440–443). Therefore, we conclude that NPF from the midgut, not the brain, has a significant role in the regulation of AKH.

5- p7 line 261 “...suggesting that...”. This conclusion seems wrong. *Bmm* expression is upregulated by *Akh>NPFRi*, but not *dHSL* expression. This suggests that *dHSL* function is not controlled (at least transcriptionally) by *NPFR* signaling in the CC. Silencing *dHSL* expression in fat cells could rescue *NPF* loss-of-function starvation sensitivity by a parallel mechanism, without implying that *dHSL* function relies on *NPFR*.

Response: Thank you for pointing out this inaccurate expression. In the revised manuscript, we have precisely described the difference in the regulatory mechanisms between *Bmm* and *dHSL* on P15, L338–339 and L345–347.

6- p12 line 289: The functional link between *NPFR* in the IPCs, and *NPF* produced from the EECs is not established. Therefore, such conclusion should not be made. Again, *NPF* could be secreted from the central brain and activate *NPFR* in the IPCs.

Response: Thank you for this important criticism. As suggested, we assessed whether *NPF* from the central brain affects *DILP* production/secretion (Extended Data Fig.15a,b). However, knockdown of brain *NPF* had no significant effect on the *DILP* mRNA and protein levels.

Therefore, we concluded that NPF from the gut, not the brain, has a significant role in the regulation of DILPs. We have mentioned this point on P19, L440–443.

7- Fig7d is not convincing. It is difficult to imagine that such a mild reduction in IPC neuronal activity could account for a notable difference in Dilp secretion.

Response: To address this question, we conducted a new experiment to measured hemolymph DILP2 level upon *NPFR* RNAi in the IPCs using the DILP2HF strain in an experiment that has been previously reported (Park et al. *PLOS Genetics* 2014; Heshan et al. *Nature Communications* 2018). The methodology is described in P32, L768–P33, L798. We found that knockdown of *NPFR* in the IPCs significantly decreased hemolymph DILP2 protein levels (Fig. 7d), consistent with reduced activity of the IPCs. These data are described on P17, L381–385.

Next, to support the reliability of the CaLexA experiment, we compared the CaLexA intensity in fed/starved conditions in both the control and *NPFR^{RNAi}* animals. Consistent with previous studies showing that starvation decreases neuronal activity of IPCs (Meschi et al. *Dev Cell*. 2019.) starvation dramatically decreased CaLexA intensity in the IPCs (Fig. 7e). Furthermore, *ad libitum* fed *NPFR^{RNAi}* animals exhibited slight decrease in the CaLexA intensity compared with *ad libitum* fed control animals (Fig. 7e). Together, these results suggest that, even if the reduction of CaLexA signal intensity is mild, knockdown of *NPFR* in the IPCs significantly reduces circulating DILP2 level. These data are described on P17, L387–393.

8- Fig7e lacks quantification and confirmation with another marker for insulin signaling. As such this piece of data is not significant.

Response: Thank you for this important suggestion. We conducted the ratiometric analysis of membrane/cytoplasmic tGPH signal and have presented the quantitative data in Fig. 7f. Based on these results, we confirmed that tGPH signalling at the plasma membrane of the fat body was significantly reduced in *Dilp2>NPFR^{RNAi}* animals.

Moreover, in response to your suggestion, we confirmed peripheral insulin signalling activity by western blotting analysis to measure the phospho-AKT/pan-AKT ratio of control (*dilp2>LacZ^{RNAi}*) and *NPFR* knockdown animals (*dilp2>NPFR^{RNAi}*). Consistent with our hypothesis, knockdown of *NPFR* in the IPCs reduced phospho-AKT/pan-AKT ratio (Fig.7g), thus, further supporting our hypothesis that *NPFR*, in IPCs, regulates insulin signalling in peripheral tissues including the fat body. These results are described on P16, L399 – P17, 402. The experimental procedure is described on P34, L806–822.

9- Throughout their manuscript, the authors present a set of related phenotypes due to EEC-specific NPF knockdown consisting in: resistance to starvation, reduced lipid accumulation, increased feeding and reduced glycemia. Experimental evidence suggest that these effects are due to NPF sending metabolic signals to the CC and the IPCs, AKH and Dilps hormones acting then as relays. However, the phenotypes observed after silencing *NPFR* in the CC and IPCs are not consistent with this idea and it is unclear what contribution *NPFR* in the IPCs and the CCs brings to the NPF knockdown phenotype. Strikingly, *Dilp2>NPFR-RNAi* animals show reduced TAG amount, glycemic levels and increased food intake, but normal resistance to starvation (Figure 8a). This contrasts with the strong decrease in resistance to starvation observed in *AKH>NPFR-RNAi* animals (Figure 5b) and indicates that this phenotype is controlled by AKT only. It also indicates that resistance to starvation is independent of decreased lipid accumulation and glycemia, and increased feeding. The authors should provide an explanation for these discrepancies and mention it in the text.

Response: As the reviewer mentioned, the hypersensitive phenotype to starvation of *NPFR* knockdown in the IPCs was mild as compared with *NPFR* knockdown in the CC. To explain the discrepancy, we hypothesized that *NPFR* knockdown in the CC might lead to significant alteration of DILP production in the IPCs. To test this hypothesis, we quantified *dilps* mRNA levels in *NPFR RNAi* in the CC. Interesting, *NPFR* knockdown in the CC decreased *dilp3* and *dilp5* mRNA levels (Extended Data Fig. 14d). In contrast, *NPFR* knockdown in the IPCs did not influence *Akh* mRNA expression (Extended Data Fig. 14e). These data suggest that *NPFR* knockdown in the CC results in not only enhancing AKH production but

also suppressing DILP production. These counterregulatory functions of AKH and DILPs may explain the differences in starvation resistance. The data are described on P178, L426 – P19, L434.

Minor remark

1- The fly food recipe used should be detailed for the reader, since many metabolic phenotypes have been shown to specifically rely on the equilibrium between food components.

Response: As the reviewer indicated, we have added the detailed composition of our fly diet in the Methods section (P25, L593–594).

Response to Reviewer #3

In this manuscript, Yoshinari et al. report that midgut-derived NPF may act as a sugar-responsive incretin-like hormone in *Drosophila*. Similar to the regulation of glucagon and insulin by incretin in mammals, the authors show that NPF is released in response to sugar and regulates the counterparts of glucagon and insulin in the fruit fly, Akh and Dilps, by suppressing the secretion of the former and promoting the secretion of the latter. NPF could be the first functional homolog of incretin identified in the invertebrate. The model proposed by the authors is very appealing and shows the deep homology in the principles of metabolic regulation between the vertebrate and invertebrate.

Although, overall, the work is interesting and significant, there are some concerns need to be addressed to make the conclusions more convincing.

Major concerns:

1, The authors showed solid data that all the phenotypes induced by knocking-out or knocking-down NPF (in Figure 1) were caused by the deficiency of NPF. However, based on the current evidence, it is not very convincing to claim that these phenotypes were caused specifically/exclusively by NPF secreted from the midgut EECs.

Considering the humoral nature of NPF, it is possible that the decrease of NPF peptides in the circulating system is sufficient to lead to these phenotypes, regardless the origin

of the decrease. For example, brain NPF secretion may also be tightly regulated by diet, in that there are also sugar/nutrient sensors in the brain. The following efforts may help address this concern.

First, test if knocking down NPF in other tissues, especially in the central nervous system, causes the same phenotypes.

Response: Thank you for raising this critical point. In response, we conducted additional experiments to knock down *NPF* specifically in the brain. To achieve this, we employed the *fbp-GAL4* driver, which is active in the NPF+ neurons in the brain but not in gut EECs (Extended Data Fig. 3a). We found that the effect of the brain-specific knockdown of *NPF* on starvation sensitivity, TAGs level, feeding amounts, DILPs level, and AKH level were not significant as compared to control animals (Extended Data Fig. 3a-e, Extended Data Fig. 15a-d). These data suggest that midgut NPF has a prominent role in suppressing lipodystrophy, which is independent from brain NPF. These results have been described on P7, L151 – P8, L157, and P19, L440–443.

Second, test if restoring NPF in other tissues, especially in the central nervous system, by either NPF overexpression in NPF neurons or optogenetic/themogenetic activation of NPF neurons, could rescue the phenotypes seen in NPF mutant. The neural-specific manipulation could be achieved by more restrictive GAL4 lines or otd-flp.

Response: In response to the reviewer's suggestion, we conducted transgenic rescue experiments in which we examined whether *NPF* expression in the brain restored the phenotype of *NPF* genetic mutant animals. To achieve this, again, we employed the *fbp-GAL4* driver. In contrast with *NPF* restoration in EECs, we found that *NPF* overexpression in the brain did not restore the hypersensitivity to starvation or the reduced TAG level in *NPF* mutant animals (Extended Data Fig. 3f, g). These data support our idea that NPF from the gut regulates lipid storage levels and has been described on P8, L157–161.

Third, simply discuss the alternative possibilities in the discussion.

Response: As per the reviewer's suggestion, we have included a

discussion regarding the difference in functions between midgut NPF and brain NPF in the Discussion section (P23, L544 – P24, L559).

2, The evidence for the connection between NPFR and IPCs neural activity is not very convincing. First, the NPFR couples to a Gi signaling pathway to inhibit NPFR-expressing neurons. Second, the quantifications in Figure 7d lack an internal control, and Figure 7e does not have quantified group data. For example, in Figure 7e, the authors could use DAPI signal as an internal control to normalize the tGPH signals. Actually, the DAPI signal in Figure 7e lower panel (NPFR-RNAi group) appears dimmer than the control groups, suggesting the change in the tGPH signals may just be an artifact due to lower background in that particular staining. In Figure 7d, the authors could also use treatments, such as starvation and re-feeding, to achieve more reliable CaLexA signals, or they could use a similar assay, the TRIC assay, which has RFP expression as an internal expression control.

Response: We appreciate the reviewer's comments. As you indicated, NPFR reportedly couples to a Gi signalling pathway. However, we would like to note that NPFR is also known to be coupled with both Gαq and Gαi subunits in heterologous expression systems, which we have described on P22, L523–525. Additionally, such dual G-protein coupling has been reported in short NPF receptor (sNPFR) signalling in the IPCs and CC (Oh et al. *Nature*. 2019). In this case, sNPFR positively regulates IPCs neuronal activity via Gαq signalling, whereas sNPFR negatively regulates the CC via Gαi signalling. Therefore, we speculate that such differential coupling in the different cell types would also occur in NPFR.

In response to the reviewer's criticism regarding the quantification of tGPH signals, we performed the ratiometric assay for membrane/cytoplasmic tGPH signals and presented the data in Fig. 7f. The quantitative data supports our conclusion that tGPH signaling at the plasma membranes of the fat body was significantly reduced in *Dilp2>NPFR^{RNAi}* animals.

Regarding an internal control and quantitative analysis of the CaLexA experiment, we have added further data in the revised manuscript. Specifically, we conducted a quantitative analysis of CaLex signals (Fig. 7e). Consistent with previous studies (Meschi et al. *Dev Cell*.

2019.), starvation was observed to reduce neuronal activity of the IPCs in both control and *Dilp2-GAL4*-driven *NPFR* RNAi animals (Fig. 7e). We also confirmed that the IPC CaLexA level of the *ad libitum* fed *dilp2>NPFR^{RNAi}* animals was stronger than that of *ad libitum* control animals (Fig. 7e). These data suggest that knockdown of *NPFR* in the IPCs slightly, however, significantly, decreases its neuronal activity. We believe that this additional data further supports our postulate that *NPFR* positively regulates IPC neuronal activity. These results are described on P17, L387–393.

Minor concerns:

1, Is there a bi-directional regulation of Akh and Dilps by NPF/NPFR? Would opposite phenotypes (in starvation resistance or TAG level) be observed if NPF/NPFR is overexpressed in a wildtype genetic background?

Response: We conducted an additional experiment to examine whether overexpression of *NPF* in the EECs affect TAG level and observed a slight increase in the TAG level (Extended Data Fig. 1g). Therefore, as you suggested, bi-directional regulation appears to exist. We have described this data on P6, L122–123.

2, The authors may include the phenotypes of *Akh>Akh-RNAi* or *Akh-A/Akh-KO* in some of the experiments in Figure 5. It would further confirm that it is Akh in the Akh cells that contributes to the phenotypes. It would also provide clues about whether the phenotypes are bi-directionally regulated.

Along the same line, the authors may test if the phenotypes of *Dilp2>Dilp2/3/5-RNAi* recapitulate the effect of knocking-down *NPFR* in IPCs to further support the model in Figure 8h.

Response: As the reviewer requested, we conducted a new experiment using *Akh>Akh^{RNAi}* animals. Consistent with the results of a previous study (Gáliková et al. *Genetics* 2015), we found that knockdown of *Akh* increased the starvation resistance and TAG abundance (Fig. 5d, f). Interestingly, whereas co-suppression of *NPFR* and *Akh* expression in the CC restored starvation sensitivity of *NPFR* knockdown, the co-suppression phenotype was stronger than a single *Akh* knockdown.

These results suggest that not only AKH, but also unknown factor(s) from the CC, contribute to the hypersensitive phenotype of the *NPFR* knockdown. We have described these points on P14, L315–318.

In response to the reviewer's second suggestion, we assessed the TAG levels in *Dilp2>Dilp2/3/5* RNAi animals. Knockdown of *dilp3* slightly decreased the TAG levels, while knockdown of *dilp2* and *dilp5* resulted in no significant differences or only a slight decreasing trend in TAG abundance, respectively (Extended Data Fig.14b). These results are consistent with a previous study showing that *dilp3* or *dilp5* mutant exhibits reduced TAG levels (Semaniuk et al. *Front. Physiol.* 2018.). Moreover, in our data, loss of NPF/NPFR signalling decreased *dilp3* and *dilp5* mRNA levels (Figure. 7b), implying that reduction of *dilp3* and *dilp5* is a cause of the attenuation of lipid storage. We have described these results on P18, L418–423.

3, The authors indicated that the behavioral and metabolic phenotypes induced by knocking-down NPFR in the IPCs is caused by attenuated insulin signaling. I think it is a bit of a stretch to attribute both phenotypes to insulin signaling. Feeding and metabolism may be regulated by different mechanisms in the IPCs. For example, most IPCs produce Dilp2/3/5 as well as DSK. It has been shown in previous studies that Dilp2>DSK-RNAi flies also show increased food intake and starvation resistance. It is not rigorous to say Dilps, regulated by NPFR, play the role in both functions without excluding the potential contribution by DSK.

Response: In response to the reviewer's comment, we examined whether *dsk* expression is influenced by knocking-down *NPFR* in IPCs. Whereas *dilp2*, *dilp3* and *dilp5* were decreased by the *NPFR* knockdown, *dsk* expression was not impacted (Extended Data Fig. 14c). In addition, *dsk* knockdown in IPCs did not decrease TAG abundance (Extended Data Fig. 14b), suggesting that NPFR in the IPCs does not affect *dsk* expression, but rather regulates TAG abundance through Dilps, not Dsk. These results are described on P18, L423–425.

4, The authors consistently observed that the expression level and the protein level of a peptide (NPF and Dilps) changed in opposite directions. It would be better if more

discussion could be made about potential mechanisms for this phenomenon. Along this line, is there any reason that the authors only showed the expression level of Akh but not the protein level?

Response: Thank you for raising this important question. In response to the reviewer's criticism, we conducted a new experiment to examine AKH protein levels in the CC cells of several genotypes. Consistent with the results for DILPs, AKH protein levels in the CC were decreased following knockdown of *NPF* or *NPFR*, while AKH expression was increased (Fig. 5a-c). These data suggest that AKH secretion is promoted in the loss of *NPF*/*NPFR* function animals. We have described these data on P13, L305 – P14, L307.

We also measured DILP2 protein levels in the hemolymph upon *NPFR* RNAi in the IPCs by using a DILP2HF strain, as previously reported (Park et al. *PLOS Genetics* 2014; Heshan et al. *Nature Communications* 2018). The methodology is described in P32, L768 – P33, L798. We found that knockdown of *NPFR* in the IPCs significantly decreased hemolymph DILP2 protein level, consistent with the reduction of DILP2 protein levels in IPCs (Fig.7d). These data are described on P17, L381–385.

Additionally, although we understand that the measurement of hemolymph *NPF* levels is important. Over the last 8 months and half, since receiving the reviewers' feedback, we have tried several experiments (ELISA, Dot blotting, western blotting, and LS-MS/MS analysis) to address this point. However, to date, we have been unsuccessful in detecting the circulating *NPF* levels in *Drosophila* hemolymph samples. We assume that this is due to the low abundance of endogenous *NPF* in the hemolymph, or its high rate of degradation in the hemolymph. In the COVID-19 pandemic in Japan, we cannot further try to promptly develop an appropriate detection method. I hope that the reviewer will understand our situation. We have, however, addressed this issue in the Discussion section of the revised manuscript (P20, L467–470). In addition, we have mentioned that we 'hypothesize' that starvation suppresses *NPF* secretion from EECs (P9, L205 – P10, L206).

Response to Reviewer #4

This manuscript describes incretin-like roles for gut-derived Neuropeptide F (NPF, homologous to mammalian NPY) in adult *Drosophila*. The same lab had previously shown that gut NPF promotes germline stem cell proliferation in mated females (PMID: 30248087) and they now explore its metabolic roles (in mated females too).

The authors show that:

1. Constitutive loss of gut enteroendocrine (EE) cell-derived NPF leads to starvation-sensitive flies. These flies have reduced triacylglyceride and a transcriptional/metabolic profile suggestive of a starvation-like state, despite eating more.
2. The levels of NPF within EE cells are sensitive to dietary sugars and dependent on the sugar transporter1 (*sut1*) gene
3. In the fed state, NPF signals through its receptor to the secretory cells that make glucagon-like Akh hormone to prevent Akh release; a compelling experiment in support of this idea is that loss of Akh can rescue the lipodystrophy resulting from NPF receptor mutation or its specific downregulation in Akh-producing cells
4. In the fed state, NPF promotes insulin release, and the NPF effects on Akh and insulin collectively regulate peripheral FOXO levels, sugar and lipid metabolism.

These conclusions are generally supported by comprehensive experiments, and the data is of very good quality. I have three general comments, and a few specific ones:

1. The roles of NPF are practically identical to those recently described for another hormone co-expressed in the same EE cells: Bursicon alpha (Bur, PMID: 30344016). This is fine, but the claims about NPF being the first incretin should be a bit toned down. More importantly, the two manuscripts together raise a few questions about how the same EE cells use different mechanisms to detect the same nutrient (sugar), resulting in potentially differential release of two different EE peptides that end up doing the same thing. This does not make sense to me and needs to be addressed somehow. Specifically, does *Sut1* regulate Burs expression? Is NPF not sensitive to *Glut1*, previously implicated in the sugar-stimulated secretion of Burs from EE cells? Is *Sut1* really acting in EE cells? The only data in support of this idea one single EE-

specific *sut1* RNAi downregulation, and *Sut1* expression in EE has not been shown. This is important because, even though the *TKg-Gal4* driver used for this downregulation is meant to be gut-specific, it is actually expressed in a small subset of neurons, so the reported phenotypes could be due to sensory/central neuronal sugar sensing/transport.

Response: In response to the reviewer's suggestion, we have removed the phrase 'incretin-like' from the title and revised the title to: 'The sugar-responsive enteroendocrine neuropeptide F regulates lipid metabolism through glucagon-like and insulin-like hormones in *Drosophila melanogaster*.' We hope that these changes will meet with the reviewer's approval. However, we have retained our idea about the incretin-like function of NPF in the Discussion section (P19, L453 – P20, L464).

As the reviewer stated, *Bursα* in the EECs is also regulated by dietary sugar and *Glut1*, whereas NPF utilizes *Sut1* to detect dietary sugar. To assess the differences in sugar-sensing mechanisms, we performed several additional experiments. First, we examined whether *Sut1* regulates *Bursα* expression by knocking down *sut1* with *TKg-GAL4*, which is also active in the *Bursα*+ EECs, and did not decrease *Bursα* expression (Extended Data Fig. 9e; P12, L261–262). Moreover, *TKg>Glut1^{RNAi}* did not affect NPF mRNA or protein levels (Extended Data Fig. 6b, c; P10, L220–224). We would also like to note that NPF and *Bursα* are produced in different regions of the midgut, namely the anterior midgut and the posterior midgut, respectively (Marianes and Spradling, *eLife* 2013). Therefore, in conjunction with our other data (Extended Data Fig. 1a-c), it is suggested that the region-specific sugar-sensing mechanism between NPF+ EECs and *Bursα*+ EECs are *Sut-1* and *Glut1*-dependent, respectively. It will also be intriguing to investigate how gut cells sense multiple dietary nutrients, in the future. We have discussed these points in the Discussion section (P20, L472 – P21, L483).

To rule out the possibility that *TKg-GAL4*+ neurons in the brain affect NPF level in the gut, we conducted brain specific *sut1* knockdown using *tub>FRT>GAL80>FRT* combined with *Otd-FLP* and *nSyb-GAL4*. We found that the brain specific *sut1* knockdown did not affect midgut NPF mRNA level, while the midgut NPF protein level was slightly reduced

(Extended Data Fig. 9a, b). Although *sut1* knockdown in the brain had a minor effect on the regulation of NPF in the EECs, *TKg>sut1^{RNAi}* had a stronger effect on NPF protein accumulation with reduced *NPF* mRNA level. Moreover, *Sut1^{KI-T2A}-GAL4* was not expressed in NPF⁺ neurons in the brain (Extended Data Fig. 9d). These results suggest that the *TKg>sut1^{RNAi}* phenotype is not a secondary effect of brain NPF. These results are described on P12, L257–261.

2. The authors make many statements about NPF secretion without directly demonstrating that NPF is, in fact, secreted as an incretin. I realise that it is not trivial to measure circulating hormones in flies, but this has been achieved for Burs made by the same cells (PMID: 30344016) using Western blots. Do the levels of circulating NPF decrease when NPF is downregulated from EEs and/or in response to sugar restriction?

Response: We appreciate your comment regarding the importance of measuring hemolymph NPF levels. However, over the last 8 months and half, since receiving the reviewers' feedback, we have tried several experiments (ELISA, Dot blotting, western blotting, and LS-MS/MS analysis) to address this point. However, to date, we have been unsuccessful in detecting the circulating NPF levels in *Drosophila* hemolymph samples. We assume that this is due to the low abundance of endogenous NPF in the hemolymph, or its high rate of degradation in the hemolymph. In the COVID-19 pandemic in Japan, we cannot further try to promptly develop an appropriate detection method. I hope that the reviewer will understand our situation. We have, however, addressed this issue in the Discussion section of the revised manuscript (P20, L467–470). In addition, we have mentioned that we 'hypothesize' that starvation suppresses NPF secretion from EECs (P9, L205 – P10, L206).

3. As far as I can tell, all genetic manipulations were performed constitutively throughout development (*TKg-Gal4* is expressed in third instar larvae possibly earlier). This makes some phenotypes difficult to interpret, such as those affecting fat body lipid and starvation sensitivity; an inability to mobilise fat stores might be expected to result in increased rather than decreased fat stores. To resolve this, the authors should test whether acute depletion of NPF in adult EE cells (by means of a *tub-Gal80ts* transgene

for example) leads to reduced lipid stores, starvation sensitivity and hyperphagia.

Response: Thank you very much for providing us with this important suggestion. In response, we conducted additional experiments to achieve the adult-specific *NPF* knockdown with *tub-GAL80^{ts}*. Consistent with the observed phenotypes of *TKg-GAL4* without *tub-GAL80^{ts}*, knockdown of *NPF* in the adult stage also exhibited reduced TAG levels, high starvation sensitivity, hypoglycaemia, and hyperphagia (Fig.1g-i, Extended Data Fig. 2a-c). These data rule out the possibility that loss of *NPF* during the larval stage affects adult metabolism. We have described these results on P6, L130 – P7, L139.

Specific comments

1. Lines 66-67. Not sure it is appropriate to describe Activin β as an enteroendocrine hormone?

Response: In response to the reviewer's comment, we have re-written this text and have described Activin- β and Bursicon- α as 'two factors that are secreted from EECs' (P4, L76–77).

2. Lines 66-68. Please provide relevant references.

Response: Thank you for bringing this to our attention. We have added relevant references (Ref #9 and #11) to this sentence. (P4, L78)

3. Line 162: “we hypothesised that starvation impairs *NPF* secretion from EECs”. Consider rephrasing: “impairs” somehow implies a failure when it is in fact the adaptive, homeostatic response.

Response: We have rephrased “impairs” to “suppresses” in response to the reviewer's comment. (P9, L205)

4. Line 214: “re-introduction” may be more accurate than “overexpression” given that the manipulation is conducted on a mutant that does not express *NPF*.

Response: We have replaced ‘overexpression’ with ‘reintroduction’ throughout the manuscript (P6, L126; P8, L157; P8, L159; P13, L292).

5. There are a few typos in some figure headings and the summary cartoon.

Response: We have fixed these typographical errors. Thank you for pointing them out.

6. The visceral muscle expression of the NPFR knockin line is surprising in light of its previously reported expression (PMID: 11897397). Although that study used larval rather than adult guts, expression appears to be in epithelial precursors/enteroendocrine cells. The two knockin lines used to assess NPFR expression also seem to differ in the CNS: one looks almost pan-neuronal which seems unlikely. Expression analysis of the endogenous NPFR transcript or protein would increase my confidence in the reported expression.

Response: We have added a comprehensive atlas of the expression pattern for *NPFR^{T2A-KI}-GAL4* in Extended Data Fig. 11 and on P12, L276 – P13, L282. We found that *NPFR^{T2A-KI}-GAL4* is expressed in the brain, CC, sNPF+ enteric neurons, Malpighian tubules, ovary, and gut. As far as we observed, two *NPFR^{KI}-GAL4* lines showed similar expression patterns in the brain, including the IPCs (Fig.7a, Extended Data Fig. 11a, 14a). We also observed expression of the *NPFR^{KI-T2A}-GAL4* line in the Malpighian tubules (Extended Data Fig. 11d), which has been referred to as the sites where *NPFR* is expressed (Chintapalli et al. *PLOS ONE*. 2012.).

More importantly, we would also like to note that two recent RNA seq studies have revealed the expression of *NPFR* in the CC (Braco et al. *bioRxiv* 2020; cited as Ref #39) and IPCs (Ryvkin et al. *Front. Behavior. Neurosci.* 2021; cited as Ref #53). The expression pattern corresponding to previous reports would support the reliability of *NPFR^{KI}-GAL4* expression. Therefore, although this previous study (Garczynski et al. *Peptides*. 2002.) did not mention *NPFR* expression in the visceral muscle, we are confident that *NPFR^{KI-T2A}-GAL4* expression in both the CC cells and IPCs is reliable. We have addressed these points in the revised manuscript on P13, L281–282 and P16, L375–376.

7. The authors may want to mention Limostatin (a fly dectetin, PMID: 25651184) in their Discussion, because together, the two studies suggest that an incretin may be regulating secretion of a dectetin.

Response: We have mentioned Limostatin on P23, L538–542 and have

discussed the possibility that Limostatin acts as a downstream factor of NPF/NPFR signalling.

8. The raw data needs to be provided for metabolomics/transcriptomics datasets.

Response: The raw transcriptome data have been deposited in the DNA Data Bank of Japan Sequence Read Archive. The data accession number is DRA010538, as described on P34, L827. In addition, we have supplied Extended Data Tables 1, 2, and 5, which represent the raw data for FPKM values used in Extended Data Fig. 4a, Extended Data Fig. 4b, and Fig. 2d, respectively.

We have also supplied Extended Data Tables 3 and 4, which represent raw data for metabolic analyses corresponding to Fig. 2a-c, and Extended Data Figure. 5a.

9. It seems interesting that NPF levels (both protein and transcript) are even higher in peptone-refed than normally fed animals. Might this suggest some integration of sugar/protein amounts by EE cells? The authors might want to discuss this.

Response: Thank you for providing us with these interesting comments. As the reviewer pointed out, peptone-refeeding increased *NPF* mRNA and protein levels compared to the normal food-fed condition (Fig. 3a-c). Although these data suggest that EECs sense sugar/protein levels in the diet, we did not identify the mechanism by which midgut *NPF* mRNA and *NPF* protein levels are upregulated by peptone feeding. In the revised manuscript, we have discussed this point in the Discussion section on P21, L489–491.

10. I believe the primary reference for the Ilp3 antibody is PMID: 18972134.

Response: We have cited the reference (Ref #91) for the anti-DILP3 antibody and anti-TK antibody on P28, L678 and P29, L682.

11. My understanding is that most experiments were done in mated females, but the CAFÉ assay used virgin females. Please clarify/justify.

Response: We used virgin female flies in this study and have described

this point in the first sentence of the subsection “Fly stock and husbandry” in the Methods section (P25, L595–596).

REVIEWERS' COMMENTS

Reviewer #2 (Remarks to the Author):

The authors have extensively and satisfactorily responded to my criticisms in this revised version. The experimental demonstration that brain NPF is not needed for the effects observed, indeed suggests that enteric NPF is involved. I acknowledge that setting up an elisa test for circulating NPF is not a simple task that can be performed in the time of revision.

Reviewer #3 (Remarks to the Author):

The authors sufficiently addressed most of my concerns in the revised manuscript. The extra experiments that they performed (during the pandemic!) make the conclusions more solid and convincing.

Reviewer #4 (Remarks to the Author):

The authors have addressed my concerns comprehensively and the revised manuscript is greatly improved. Specifically, they have convincingly ruled out possible developmental and neuronal effects of NPF. The new Sut1 tools (especially the Sut1 knock-in line) and new experiments with a glucose sensor provide compelling evidence for Sut1 acting as an important regulator of glucose intracellular levels in enteroendocrine cells. The authors have also clarified some of the issues regarding Burs/NPF co-release from the same cells, and rectified inaccuracies/overstatements in the manuscript.

Collectively, the revised manuscript provides a very comprehensive description of an incretin-like pathway that couples sugar sensing in the gut to release of insulin and glucagon-like hormones. Congratulations!